# Puf6 primes 60S pre-ribosome nuclear export at low temperature

Stefan Gerhardy[1,2,9,12], Michaela Oborská-Oplová [1,2,12], Ludovic Gillet [3], Richard Börner [4,10],
Rob van Nues [5,6], Alexander Leitner [3], Erich Michel[7], Janusz J. Petkowski [2,11], Sander Granneman[5],
Roland K. O. Sigel [4], Ruedi Aebersold [3,8] & Vikram Govind Panse [1,8✉]

Productive ribosomal RNA (rRNA) compaction during ribosome assembly necessitates establishing correct tertiary contacts between distant secondary structure elements. Here, we quantify the response of the yeast proteome to low temperature (LT), a condition where aberrant mis-paired RNA folding intermediates accumulate. We show that, at LT, yeast cells globally boost production of their ribosome assembly machinery. We find that the LT-induced assembly factor, Puf6, binds to the nascent catalytic RNA-rich subunit interface within the 60S pre-ribosome, at a site that eventually loads the nuclear export apparatus. Ensemble Förster resonance energy transfer studies show that Puf6 mimics the role of $Mg^{2+}$ to usher a unique long-range tertiary contact to compact rRNA. At LT, puf6 mutants accumulate 60S pre-ribosomes in the nucleus, thus unveiling Puf6-mediated rRNA compaction as a critical temperature-regulated rescue mechanism that counters rRNA misfolding to prime export competence.

[1] Institute of Medical Microbiology, University of Zurich, Zurich, Switzerland. [2] Institute of Biochemistry, ETH Zurich, Zurich, Switzerland. [3] Institute of Molecular Systems Biology, ETH Zurich, Zurich, Switzerland. [4] Department of Chemistry, University of Zurich, Zurich, Switzerland. [5] Centre for Synthetic and Systems Biology (SynthSys), University of Edinburgh, Edinburgh, UK. [6] Institute of Cell Biology, University of Edinburgh, Edinburgh, UK. [7] Department of Biochemistry, University of Zurich, Zurich, Switzerland. [8] Faculty of Science, University of Zurich, Zurich, Switzerland. [9] Present address: Department of Early Discovery Biochemistry, Genentech Inc., South San Francisco, CA, USA. [10] Present address: Laserinstitut Hochschule Mittweida, University of Applied Sciences Mittweida, Mittweida, Germany. [11] Present address: Department of Earth, Atmospheric, and Planetary Sciences, Massachusetts Institute of Technology, Cambridge, MA, USA. [12] These authors contributed equally: Stefan Gerhardy, Michaela Oborská-Oplová. ✉email: vpanse@imm.uzh.ch

Most RNAs exert their function after folding into their native 3D structures. However, intramolecular sequence complementarities make RNA-folding processes promiscuous, thus delaying the acquisition of functionality. Low temperature exacerbates RNA-misfolding due to thermodynamic stabilization of trapped non-native states. To overcome these barriers, cells deploy RNA-chaperones and RNA-helicases that prevent and/or resolve aberrant RNA folding intermediates[1].

The ribosome is one of the most abundant RNA:protein complexes assembled by incorporating ribosomal proteins (r-proteins) into dynamically folding ribosomal RNA (rRNA). In eukaryotes, the large 60S subunit contains three rRNAs (25S, 5.8S, 5S) and 46 r-proteins, whereas the small 40S subunit contains a single rRNA (18) and 33 r-proteins[2]. Eukaryotic ribosome assembly is initiated co-transcriptionally in the nucleolus by RNA polymerase-I driven transcription of rDNA repeats. The emerging pre-rRNA associates with U3 snoRNP, 40S-specific r-proteins, and ~100 assembly factors to form the small subunit precursor[3]. Cryogenic electron microscopy (cryo-EM) studies suggest that assembly factors and U3 snoRNP encapsulate and guide hierarchical 5′-to-3′-oriented pre-rRNA folding[4]. Following endonucleolytic cleavage and release of the 40S pre-ribosome, the growing 27S pre-rRNA associates with 60S-specific r-proteins and assembly factors to concomitantly fold rRNA domains (I to VI) to form the 60S pre-ribosome. Cryo-EM studies have revealed that the 5′ rRNA domains I and II within 27S pre-rRNA fold to a near-mature conformation and interact with the 3′ terminal rRNA domain VI, to form the characteristic arch-like solvent-exposed backside of the 60S subunit[5]. The internal transcribed spacer 2 (ITS2) located between 5.8S and 25S rRNA within 27S pre-rRNA and associated assembly factors forms the "foot" structure already visible in these early states. However, the central rRNA domains III, IV, and V, which form the catalytic subunit interface remain unresolved. Structural studies implicate non-linear compaction wherein the near-native rRNA domains I, II, and VI establish tertiary contacts with the flexible rRNA domains III, IV, and V[5,6]. Mechanisms that drive the correct formation of these long-range contacts during 27S pre-rRNA folding remain unknown.

Establishing correct tertiary contacts within 27S pre-rRNA is critical for polypeptide exit tunnel formation, productive pre-rRNA processing of ITS2, and the incorporation/rotation of the 5S RNP[5,6]. These interactions stabilize the rRNA-rich subunit interface and provide a binding platform for the GTPase Nog2 and the assembly factor Sda1 which recruits the Rix1-Ipi-complex and the AAA-ATPase Rea1 to the 60S pre-ribosome[7,8]. The coordinated release of Rea1, the Rix1-Ipi-complex, and Nog2 functions as a checkpoint that prevents premature recruitment of the export adaptor Nmd3 for Crm1-mediated transport into the cytoplasm[9].

Nucleolar ribosome biogenesis steps require coordinated rearrangements of RNA:RNA and RNA:protein interactions that enable hierarchical RNA folding, and incorporation of assembly factors and r-proteins[10]. The multitude of possible intramolecular base-pairings and the increased stability of mispaired structural elements pose a formidable barrier for correct pre-rRNA folding, especially at low temperatures[11–13]. Here, we measured how the yeast proteome adapts to low temperatures (LT). We found that yeast cells boost assembly factor production at LT. We show that the 60S assembly factor, Puf6, is induced at LT to usher a non-canonical tertiary contact between distant secondary structure elements within the catalytic rRNA-rich subunit interface important for recruiting the nuclear export machinery. Our findings provide insights into how Puf6-mediated rRNA compaction at LT primes 60S pre-ribosome nuclear export.

## Results

**Yeast cells boost the production of ribosome assembly factors at low temperature.** To compare proteomes of yeast cells grown at different temperatures, we employed sequential window acquisition of all theoretical fragment ion spectra (SWATH) mass spectrometry (MS)[14]. SWATH-MS is an effective approach that has reliably quantified proteomes of diverse model organisms, and tissues from disease states[15].

We subjected lysates derived from WT yeast cells grown at 20, 25, 30, and 37 °C in rich media to SWATH-MS (Fig. 1a). A total of 2821 proteins were detected and their relative abundances at different temperatures were quantified. By employing hierarchical clustering algorithms, we grouped the proteome profiles of individual proteins that showed a more than 1.5-fold change in levels across the different temperatures. Based on these analyses, 426 proteins were organized into nine mutually exclusive clusters (Fig. 1b). We characterized the individual members within each cluster based on their functional GO terminology using the pathway enrichment analysis tool DAVID (Supplementary Data 1; http://david.abcc.ncifcrf.gov/)[16]. Components of clusters 5–8 showed increased protein abundance levels with increasing temperatures. These clusters enriched 15 heat shock proteins (HSPs) including members of the HSP70 and the HSP90 families as well as small HSPs and their regulators that are involved in the cellular response to heat stress. Clusters 9 and 3 enriched proteins devoted to different aspects of cellular metabolism including oxidative phosphorylation, and purine/pyrimidine metabolism, respectively (Supplementary Data 1).

Factors within clusters 1, 2, and 4 showed increased levels at 20 °C (hereafter referred to as low temperature, LT), in comparison to levels at 30 °C. Strong overrepresentation of ribosome assembly factors (1.09E−09, cluster 2) and rRNA processing factors (3.97E−11, cluster 2) was observed in these clusters (Fig. 1c and Supplementary Data 1). Several identified factors are organized as sub-complexes that scaffold pre-rRNA on maturing pre-ribosomes (Supplementary Figs. 2 and 3). For example, the $A_3$ factors, Nop15, Cic1, Rlp7, Nop7, and N-terminus of Erb1 (Fig. 1c, left panel and Supplementary Fig. 2, in yellow) stabilize ITS2 within the 27S pre-rRNA during 60S assembly[5,6,17]. These factors are essential for ITS2 processing and form the "foot structure", a structural landmark on early 60S pre-ribosomes[17,18]. Similarly, the ribosomal-like protein Mrt4 (Fig. 1c, left panel and Supplementary Fig. 3, light green), which scaffolds folding of the rRNA base of the P-stalk during nuclear maturation, and its cytoplasmic release factor Yvh1[19,20] showed increased abundance at LT. Rpf2 and the AAA-ATPase Rea1 (Fig. 1c, left panel and Supplementary Fig. 2, red) were induced at LT; Rpf2 and Rea1 are involved in the process of 5S RNP docking and rotation on the pre-ribosome and maturation of the central protuberance, another structural landmark of the 60S subunit[7,21]. Interestingly, several r-proteins located on the solvent-exposed side of the 60S subunit were induced at LT (Fig. 1c, right panel, purple). Finally, several assembly factors (Fig. 1c, left panel, orange and dark red) whose location on maturing pre-ribosomes remain unknown were induced at LT. A (post-hoc) t-test of the protein abundances between 37 °C and 20 °C confirmed that 43 out of the 54 identified assembly factors and r-proteins showed a significant increase in abundance (adjusted p-value < 0.05) with a median fold change of 2.5. We conclude that yeast cells globally boost the production of ribosome assembly factors and r-proteins at LT.

**Puf6 induction at LT ensures efficient 60S pre-ribosome export.** Cryo-EM studies are pinpointing the RNA-binding sites of assembly factors on 60S pre-ribosomes. Yet, the locations of

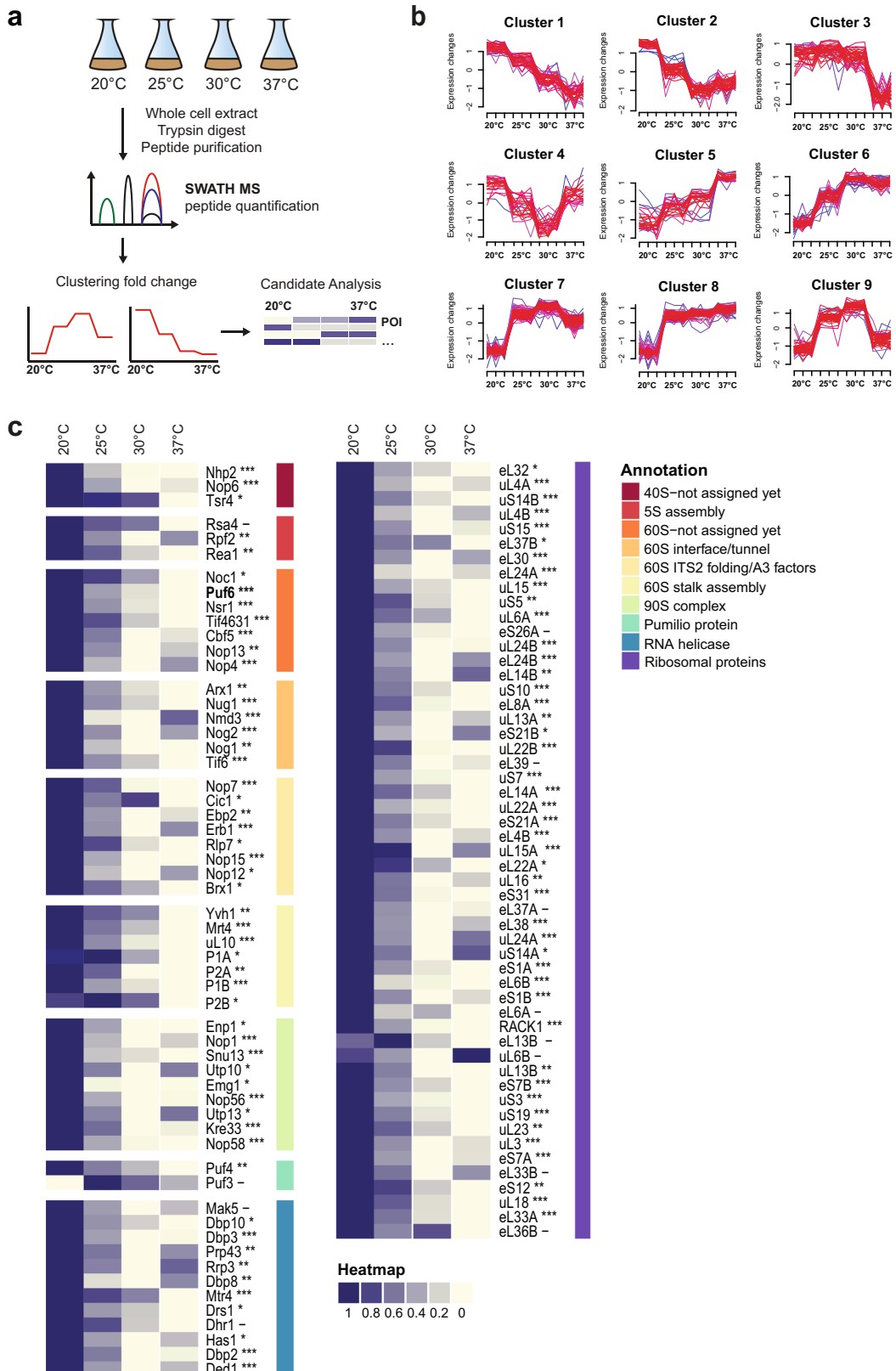

several factors especially those associated with rRNA domains III, IV, and V during early nucleolar maturation remain unknown[8,22]. One such factor is Puf6, which we found to be induced at LT (Fig. 1c, left panel, orange bar). In agreement with the SWATH-MS data, Western analyses showed a significant increase in Puf6 protein levels when yeast cells were grown at

20 °C, as compared to 37 °C (Fig. 2a). Puf6 protein levels were induced when yeast cells growing at 37 °C were shifted to 20 °C. Conversely, Puf6 protein levels decreased when yeast cells growing at 20 °C were shifted to 37 °C (Fig. 2b).

Puf6 is a conserved assembly factor that contains an atypical Pumilio repeat RNA-binding domain[23–25]. Puf6-deficient yeast

**Fig. 1 Quantifying alterations in the proteome by SWATH-MS in yeast cells grown at different temperatures. a** Experimental workflow for SWATH-MS characterization of proteomes of yeast cells grown at 20, 25, 30, and 37 °C. Whole-cell extracts derived from WT yeast cells grown at indicated temperatures were subjected to the SWATH-MS and hierarchical clustering of proteins based on their abundance profile at different temperatures was performed. **b** Hierarchical clustering profiles of protein levels at the indicated temperatures. **c** Relative enrichment of ribosome assembly factors and ribosomal proteins from clusters 1, 2, and 4. Maximum enrichment is depicted in purple and minimum enrichment in gold. Proteins were further grouped based on their GO annotations—according to the indicated color code. Significance of the protein change was assessed using a one-way ANOVA test, with the resulting $p$-value further adjusted using the FDR method; ***adj.p.value < 0.0001; **adj.p.value < 0.001; *adj.p.value < 0.01; −adj.p.value ≥ 0.01.

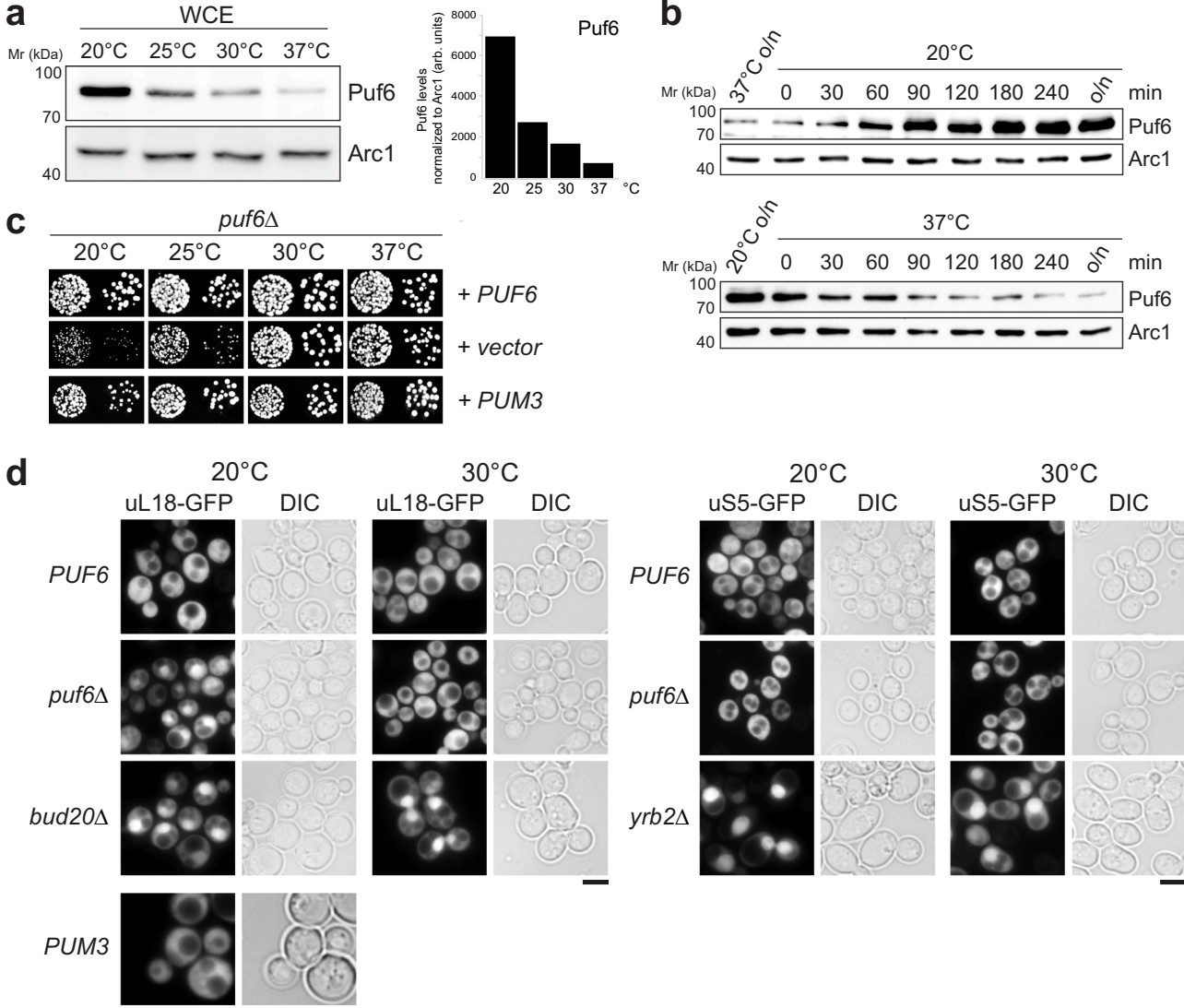

**Fig. 2 Puf6 is required for 60S pre-ribosome export at LT. a** Whole-cell extracts (WCE) from WT yeast cells grown at indicated temperatures were separated by SDS-PAGE and analyzed by Western blotting using a Puf6-specific antibody. Arc1 was used as a loading control. Protein levels were quantified by ImageJ software. The right panel depicts Puf6 protein levels normalized to the loading control Arc1. arb. units stand for arbitrary units. **b** Whole-cell extracts were prepared from WT yeast cells grown at 37 °C, and then shifted to 20 °C or vice versa for the indicated times were separated by SDS-PAGE and analyzed by Western blotting with a Puf6-specific antibody. Arc1 was used as a loading control. **c** *puf6Δ* cells expressing WT-Puf6 and human Pum3 were spotted in serial tenfold dilutions on selective minimal medium plates and grown at indicated temperatures for 3–5 days. **d** The indicated strains expressing uL18-GFP and uS5-GFP were grown to mid-log phase at 20 °C and at 30 °C. Localization of the reporters was visualized by fluorescence microscopy. *bud20Δ* and *yrb2Δ* cells expressing uL18-GFP and uS5-GFP, respectively, were used as positive controls. Scale bars = 5 μm. Source data are provided as a Source Data file.

cells accumulate 27S and 7S pre-rRNA and exhibit reduced levels of 60S subunits at 20 °C[23]. Puf6 has been implicated in the loading of the r-protein eL43 during 60S assembly[26,27]. We initiated functional studies by disrupting the endogenous *PUF6* gene encoded by the open reading frame in WT diploid yeast

cells. Tetrad analysis yielded two spores with WT growth rates and two spores with a slow-growth phenotype at 25 °C, which carry the *PUF6* deletion (*puf6Δ*). In agreement with previous studies[23], we found that growth of the *puf6Δ* mutant is impaired at 20 and 25 °C, as determined by the size of single colonies. At 30

and 37 °C, the *puf6Δ* mutant grew nearly like WT cells (Fig. 2c). Next, we examined the localization of 40S and 60S subunits in the *puf6Δ* mutant at 20 and 30 °C. At 20 and 30 °C, WT cells showed cytoplasmic localization for both 40S and 60S subunit reporters, uS5-GFP and uL18-GFP, respectively. In contrast, *puf6Δ* cells revealed a nuclear accumulation of uL18-GFP at 20 °C, but not at 30 °C. Localization of the uS5-GFP remained unaffected in *puf6Δ* cells at 20 and 30 °C (Fig. 2d). Thus, the slow growth of the *puf6Δ* mutant at 20 °C correlates with impaired 60S pre-ribosome export. Impaired growth of the *puf6Δ* mutant at 20 °C, and increased Puf6 protein levels in WT cells at 20 °C point to a requirement of Puf6 during 60S assembly at LT. Expression of the human ortholog, Pum3 (also known as Puf-A or KIA0020), rescued the slow growth and 60S pre-ribosome export defect of the *puf6Δ* mutant at 20 °C, indicating a conserved role during 60S assembly (Fig. 2c, d).

**Puf6 co-enriches with early nucleolar 60S pre-ribosomes**. Previous studies have suggested a role for Puf6 in the localization and translational repression of specific mRNAs[28–30]. However, in agreement with cell-biological studies[26,31], we found that Puf6-GFP predominantly co-localizes with Gar1-mCherry, an established nucleolar marker (Fig. 3a). To test whether Puf6 shuttles between the nucleus and cytoplasm, we employed the established heterokaryon assay[32]. A strain expressing a Puf6-GFP was mated with a *kar1-1* expressing strain, in which mating and cell conjugation are not followed by nuclear fusion, leading to heterokaryon formation. In order to distinguish the two nuclei in the resulting heterokaryon, the nuclear pore protein Nup82 was tagged with mCherry in the *kar1-1* strain. As controls, we used the shuttling Arx1-GFP and non-shuttling Gar1–GFP strains, respectively. While Arx1-GFP localized to both nuclei after mating, Puf6-GFP and Gar1–GFP were never observed in the nucleus of the *kar1-1* strain (Fig. 3b). While these data are consistent with an exclusively nuclear location of Puf6 in vivo, a small undetectable cytoplasmic fraction of Puf6 may repress translation.

We investigated the maturation stage at which Puf6 is recruited to and released from the 60S pre-ribosome[33]. For this, we purified 60S pre-ribosomes at distinct maturation stages using different TAP-tagged bait proteins: Ssf1-TAP purifies an early nucleolar 60S pre-ribosome; Rix1-TAP purifies a nucleoplasmic 60S pre-ribosome; Arx1-TAP purifies a late export competent 60S pre-ribosome; and Kre35-TAP purifies an exclusively cytoplasmic 60S pre-ribosome. Western analyses revealed that Puf6 co-enriches mainly with the nucleolar Ssf1-TAP particle and to a lesser extent with the nucleoplasmic Rix1-TAP. Puf6 did not co-enrich with Arx1-TAP and Kre35-TAP (Fig. 3c). Consistent with this, we found that the Puf6-TAP co-enriched assembly factors such as Rrp5, Noc1, Rrp1, Urb1, Mak11, etc. that co-enrich with nucleolar 60S pre-ribosomes (Fig. 3d). We suggest that Puf6 exerts its function during early nucleolar 60S assembly.

**The Pumilio repeat domain recruits Puf6 to the 60S pre-ribosome**. Puf6 belongs to the Pumilio and fem3 mRNA-binding factor (PUF) family, a large eukaryotic family of RNA-binding proteins. However, within the seven members of the yeast PUF family, Puf6 is the most diverged member[34] and exhibits a different domain organization (Fig. 4a). In addition to the C-terminal RNA-binding PUF repeats, the human ortholog Pum3/Puf-A contains an N-terminal extension with three additional PUF repeats[35], which are also found in yeast Puf6 (Fig. 4a). Homology modeling and small-angle X-ray scattering (SAXS) indicated the Puf6 structure resembles the L-shaped conformation Puf-A[35]. Using Pum3/Puf-A (PDB 4WZW) as a template, we generated homology models for yeast Puf6 (residues 118–654)

that permitted structure-guided studies (Fig. 4b). We substituted several residues in Puf6 to glutamic acid to induce repulsion with the phosphate backbone of the RNA substrate, and impair Puf6:RNA interactions (Supplementary Fig. 4b, c, d). R172 and Y208 are part of a basic patch shown to be critical for correct 7S pre-rRNA processing and localization of ASH1 mRNA[35] and correspond to R181, N217, and K440 in human Pum3 respectively (Fig. 4c). While single mutants showed a minor growth defect at 20 °C (Supplementary Fig. 4b), mutants in repeat 1 (R172E) or repeat 2 (Y208E) when combined with the repeat 5 (R431E) induced a strong growth impairment (Fig. 4d). Like the *puf6Δ* strain, the puf6 $^{R172E\ R431E}$ and puf6 $^{Y208E\ R431E}$ mutants accumulated uL18-GFP in the nucleus at 20 and 25 °C, but not at 30 and 37 °C (Fig. 4d, e). Both mutant proteins did not co-enrich with 60S pre-ribosomes (Fig. 4f). This was not due to altered stability and/or localization as Western analyses of whole-cell extracts (WCE), and fluorescence microscopy, revealed that their protein levels and the nucleolar signal were indistinguishable from wild-type Puf6 (Fig. 4f, WCE lower panel, g). We conclude that the PUF repeats target Puf6 to the 60S pre-ribosome.

**Puf6 binds to H68 rRNA within a 60S pre-ribosome**. Proteomic analyses indicated that Puf6 is co-transcriptionally recruited to rRNA domain IV within 27S pre-rRNA[36,37]. However, the RNA-binding site of Puf6 on the 60S pre-ribosome remained unclear. We employed ultraviolet cross-linking and analysis of complementary cDNA (CRAC)[38] to localize its RNA-binding site within the 60S pre-ribosome. For this, we generated a yeast strain wherein Puf6 was fused with a C-terminal His₆-ProteinA tag at the genomic locus. In vivo cross-linked RNA:Puf6-His₆-ProteinA complexes were purified under denaturing conditions, and the recovered RNA fragments were amplified after linker ligation and cross-linked sites were mapped by deep sequencing. While specific hits were identified within the ITS2 sequence, the major Puf6 contact sites mapped to helix 66, 68, and 69 (hereafter referred to as H66, H68, and H69) within the rRNA domain IV of 25S rRNA (Fig. 5a). These secondary structure elements are present at the RNA-rich subunit interface and line the catalytic centers on the 60S subunit[2]. During 60S assembly, H66, H68, and H69 provide a binding platform for Nog2, a placeholder for the export adaptor Nmd3 (Fig. 5a, right panel)[9]. During translation, these rRNA helices make contact with the 40S subunit and restrict the tRNA corridor at the P- and E-site.

To validate the CRAC data, we employed yeast three-hybrid and cross-linking mass spectrometry (XL-MS). Yeast three-hybrid studies revealed a strong interaction between Puf6 and H68, but not H38, H66, or H69 present within 25S rRNA (Fig. 5b). Moreover, both PUF repeat mutant proteins, puf6 $^{R172E\ R431E}$ and puf6 $^{Y208E\ R431E}$, that were not recruited to the 60S pre-ribosomes in vivo (Fig. 4f) did not interact with H68 as judged by yeast three-hybrid studies (Fig. 5b). Given that H68 is solvent-exposed at the rRNA-rich subunit interface, we tested whether recombinant Puf6 can bind to a mature 60S subunit. Sedimentation binding assays confirmed a robust interaction between Puf6 and a 60S subunit (Fig. 5c). Saturation of the 60S-bound assembly factor with increasing Puf6 concentrations, and a strong impairment in the binding of recombinant PUF repeat mutants to the 60S subunit confirmed the specificity of the interaction (Fig. 5c, d). We performed crosslinking-mass spectrometry (XL-MS) analyses of a reconstituted 60S:Puf6 complex using disuccinimidyl suberate (DSS) that can covalently link lysine residues that are up to 37 Å apart[39,40]. These analyses revealed protein:protein cross-links between Puf6 and the r-protein uL2 (Fig. 5e and Supplementary Data 2). uL2 is located at the subunit interface of the 60S, where it contacts H66–H68 within rRNA domain IV. All identified cross-

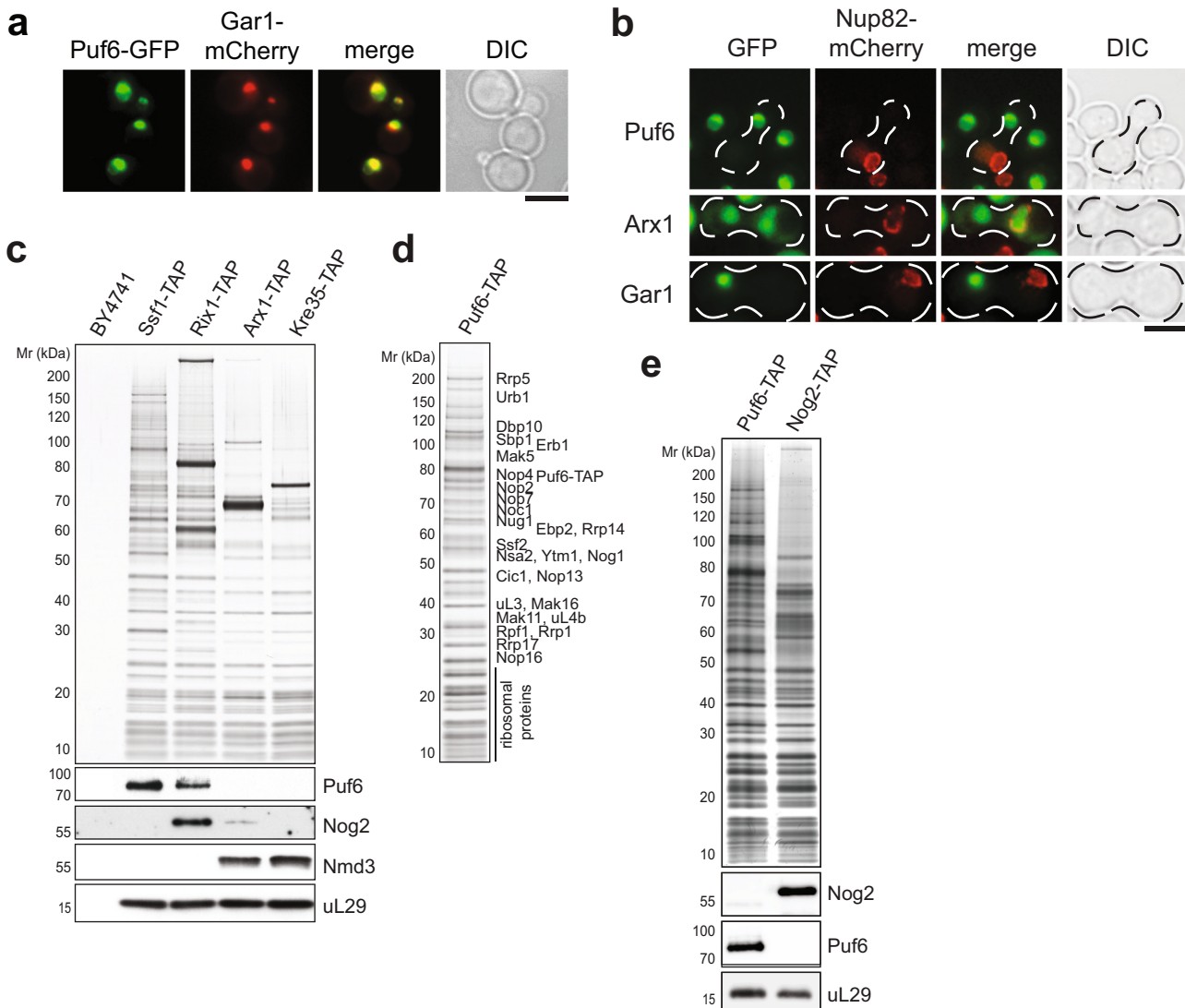

**Fig. 3 Puf6 is enriched in the nucleolus and associates with early 60S pre-ribosomes. a** Yeast cells expressing Puf6-GFP and the nucleolar marker Gar1-mCherry were grown to mid-log phase in YPD and then analyzed by fluorescence microscopy. Scale bar = 5 μm. **b** Cells expressing Puf6-GFP, Arx1-GFP, or Gar1–GFP were mated with a kar1-1 strain expressing Nup82-mCherry. The resulting heterokaryons were analyzed by fluorescence microscopy. Arx1-GFP and Gar1–GFP serves as a positive and negative control for the shuttling assay, respectively. Scale bar = 5 μm. **c** TAP-eluates of the early to late 60S pre-ribosomes isolated via the indicated TAP-baits were separated on a NuPAGE 4–12% Bis-Tris gradient gel and subjected to Silver staining or Western blotting using antibodies directed against Puf6, Nog2, and Nmd3. The r-protein uL29 (yeast Rpl35) served as a loading control. WT-BY4741 (untagged strain) served as the negative control. **d** The Puf6-TAP particle was separated on a NuPAGE 4–12% Bis-Tris gradient gel and analyzed by Silver staining. The indicated proteins were identified by mass spectrometry. **e** Puf6-TAP and Nog2-TAP eluates were separated on a NuPAGE 4–12% Bis-Tris gradient gel and subjected to Western blotting using the indicated antibodies. The r-protein uL29 (yeast Rpl35) served as a loading control. Source data are provided as a Source Data file.

links involve lysine residues located within the N- and C-terminal extensions flanking the Puf6-Pumilio repeat domain (Fig. 5e).

The hairpin-loop of H68 contains a GAAA-tetraloop motif and engages in a non-canonical tertiary contact with a kissing loop on the mature 60S subunit. *E. coli* H68, however, does not form such a tertiary contact within the 50S subunit[41] and does not interact with Puf6 as judged by yeast three-hybrid studies (Fig. 5f). We generated chimeric constructs between *E. coli* and yeast H68 to dissect the H68:Puf6 interaction by yeast three-hybrid studies. In one mutant, we swapped the hairpin-stem of yeast H68 with the hairpin-stem from *E. coli* H68, thus retaining the yeast H68 hairpin-loop region (H68 Tip$_{Sc}$). Conversely, we swapped the yeast H68 hairpin-loop region with the *E. coli* H68 hairpin-loop, thus retaining yeast H68 rRNA hairpin-stem (H68 Tip$_{E. coli}$; Fig. 5a). We found that Puf6 interacted

with the chimeric RNA that contained the yeast H68 hairpin-loop region (Fig. 5f). These analyses are consistent with CRAC studies which revealed a major cross-link between Puf6 and U2217 located very close to the GAAA-tetraloop motif within H68 (Fig. 5a).

Based on CRAC, yeast three-hybrid, and XL-MS analyses, we suggest that Puf6 binds the 60S pre-ribosome through interactions between the Puf6-Pumilio repeat domain and the H68 hairpin-loop region.

**Puf6 removal from the 60S pre-ribosome precedes Nog2 recruitment.** In addition to their critical role during translation, H68 and H69 within RNA domain IV of 25S rRNA provide a binding platform for Nog2, a placeholder that prevents

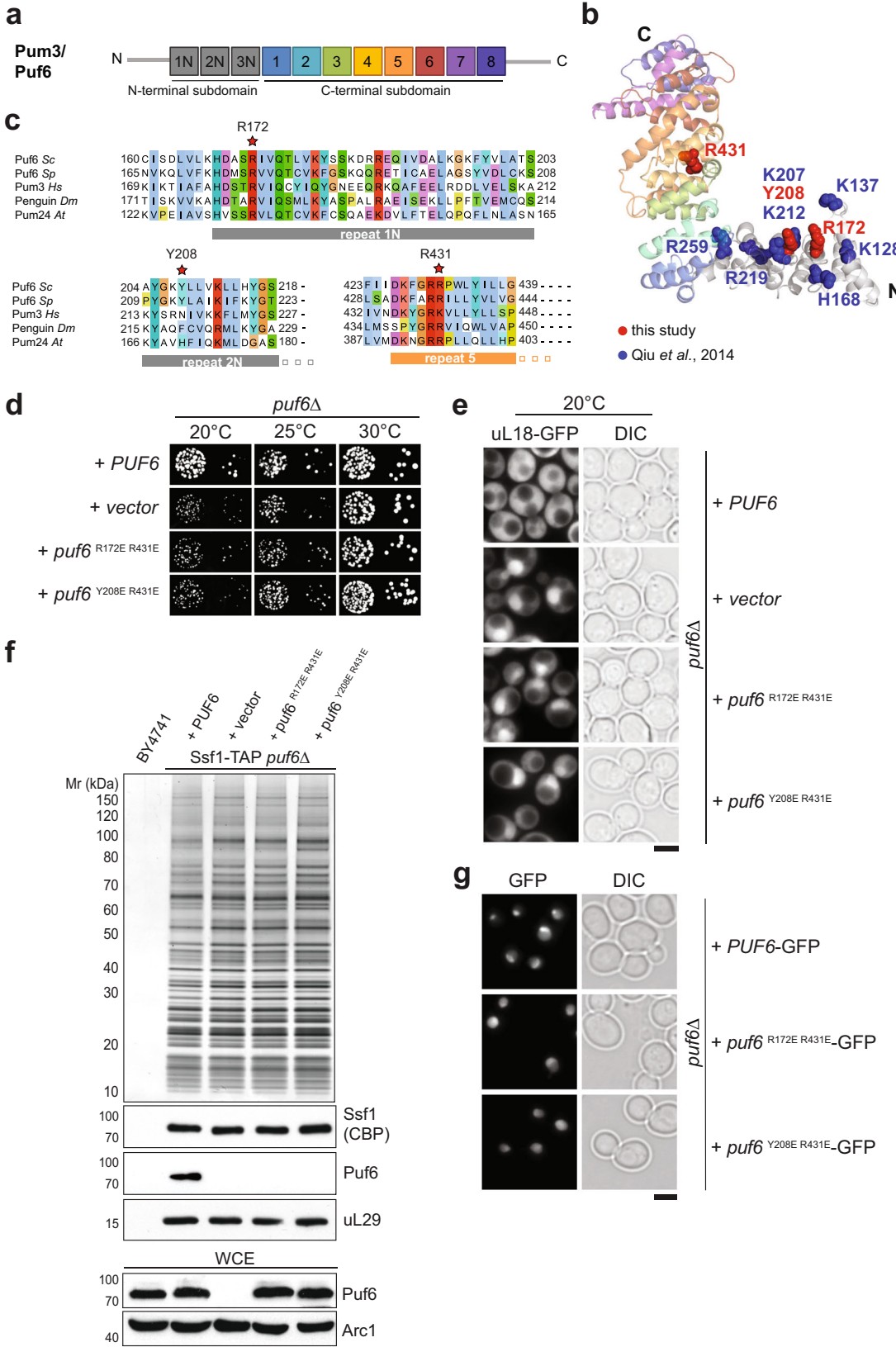

premature recruitment of Nmd3, until late nuclear stages of 60S assembly[9]. Given that Puf6, Nog2 and Nmd3 share a common binding site (Fig. 5a, right panel) we investigated their order of recruitment to the 60S pre-ribosome. We monitored the co-enrichment of Puf6, Nog2, and Nmd3 with 60S pre-ribosomes representing distinct maturation stages by Western blotting

(Fig. 3c). Puf6 co-enriches mainly with the nucleolar Ssf1-TAP, but lower levels of Puf6 were detected on the nucleoplasmic Rix1-TAP particle (Fig. 3c). On the other hand, Nog2 is not found on the Ssf1-TAP but co-enriched only with the Rix1-TAP particle. Nog2 is hardly detected on the export competent Arx1-TAP particle (Fig. 3c). Further, we found that Nog2-TAP did

**Fig. 4 The Pumilio repeat domain recruits Puf6 to 60S pre-ribosomes. a** Repeat organization of an atypical Pumilio repeat domain-containing Pum3/Puf6. **b** Cartoon representation of Puf6 homology model, visualized by PyMoL. The structure modeling of Puf6 was performed by using PHYRE2 server[77]. Of the 656 residues of Puf6, 118–654 residues could be modeled with 100% confidence using Pum3/Puf-A crystal structure (PDB ID: 4WZW) as the preferred template. Substituted residues within this study are shown in red. Previously substituted residues with altered affinities to nucleic acid substrates are shown in blue[35]. **c** Partial sequence alignment of Puf6—non-canonical PUF-repeat 1, 2, and canonical repeat 5 from *Saccharomyces cerevisiae* (*Sc*), *Schizosaccharomyces pombe* (*Sp*), *Homo sapiens* (*Hs*), *Drosophila melanogaster* (*Dm*) and *Arabidopsis thaliana* (*At*). The residues indicated by a red asterisk were substituted in yeast Puf6 for functional studies: R172, Y208, and R431 corresponding to R181, N217, and K440 in human Pum3/Puf-A[35] **d** *puf6Δ* cells expressing WT-Puf6, and the indicated puf6 mutants were spotted in serial 10-fold dilutions on selective minimal medium plates and grown at indicated temperatures for 3–5 days. **e** The indicated strains expressing the 60S reporter, uL18-GFP, were grown at 20 °C till mid-log phase. Localization of uL18-GFP was analysed by fluorescence microscopy. Scale bar = 5 µm. **f** Ssf1-TAP was isolated from *PUF6, puf6Δ* and the indicated puf6 mutants. The TAP eluates were separated on a NuPAGE 4–12% Bis-Tris gradient gel and analyzed by Silver staining and Western blotting using antibodies directed against the bait (Ssf1-CBP) and Puf6. The r-protein uL29 (yeast Rpl35) was used as a loading control. Prior to TAP, whole-cell extracts (WCE) from the indicated yeast strains were separated by SDS-PAGE and analyzed by Western blotting using a Puf6-specific antibody. Arc1 was used as a loading control. **g** Yeast cells expressing C-terminally GFP tagged variants of Puf6 and puf6 mutants were grown to mid-log phase and analysed by fluorescence microscopy. Scale bar = 5 µm. Source data are provided as a Source Data file.

not co-enrich Puf6, and conversely, Puf6-TAP did not co-enrich Nog2 (Fig. 3e). These data are consistent with the notion that Puf6 removal from the 60S pre-ribosome precedes Nog2 recruitment. We suggest that Rix1-TAP purifies a state of a 60S pre-ribosome that is loaded with Puf6, and a later state wherein Puf6 has been evicted and Nog2 has been recruited.

**Puf6 promotes an unusual tertiary contact within the 60S pre-ribosome.** Within a mature 60S subunit, H68 rRNA engages in a non-canonical tertiary contact at the subunit interface that is conserved in all eukaryotes (Fig. 6a)[2,42]. This long-range contact involves a GAAA-tetraloop motif ($TL_{GAAA}$) within the hairpin-loop of H68 and a kissing loop (KL) formed between H22 and H88. In early nucleolar states, H22 within rRNA domain I is bound to assembly factors Ebp2 and Brx1 presumably to prevent premature KL formation (Fig. 6b, State 3, left panel)[6]. The rRNA domains IV and V that contain H68 and H88, respectively, are not fully resolved in these states (Fig. 6b)[5,6]. We wondered whether Puf6 binding prevents H68 $TL_{GAAA}$ to prematurely interact with the KL and/or undergo non-productive interactions with other tetraloop receptors within the pre-rRNA. To test this idea, we developed a minimal system using the tip region of H68, H88, and H22 (Fig. 6a, c and Supplementary Table 3) for Förster resonance energy transfer (FRET) experiments. KL formation between H22 and H88 (KL construct) and docking of the $TL_{GAAA}$ in H68 onto the KL (KL-$TL_{GAAA}$ construct) in vitro was monitored by FRET using two RNA constructs comprising the same sequence and a sCy3 donor position but differing in the position of sCy5 acceptor dyes (Fig. 6c and Supplementary Table 3). The proximity of dyes on the RNA constructs reports either KL formation (Fig. 6c, KL) or KL-$TL_{GAAA}$ docking (Fig. 6c, KL-$TL_{GAAA}$). A knowledge-based structural model generated using RNA-Composer[43] matched well with the structure of H22, H68, and H88 ensemble within the 60S subunit with an RMSD of 1.6 Å (Fig. 6a, right panel).

RNA folding is sensitive to mono- and divalent cations[44]. The monovalent $K^+$ shields the negatively charged phosphate backbone and facilitates RNA folding into secondary structure elements. The divalent $Mg^{2+}$ effectively stabilizes the high charge density during the process of binding and/or docking of distant RNA structure elements. $Mg^{2+}$ alleviates the electrostatic stress arising from closely packed phosphates to a similar extent as two $K^+$ ions, but at a lower entropic cost because fewer ions are confined near the RNA[11,45,46]. We found a low FRET efficiency at low $K^+$ concentrations for both KL and KL-$TL_{GAAA}$ reporters indicating that these structures are not populated (Fig. 6c, left panel; d, compare white bars of KL and KL-$TL_{GAAA}$). Titration with increasing $K^+$ concentrations revealed a corresponding

increase in FRET efficiency for KL and KL-$TL_{GAAA}$ reporters (Fig. 6c, left panel; d, compare blue bars of KL and KL-$TL_{GAAA}$). A higher FRET efficiency was observed for the KL reporter as compared to the KL-$TL_{GAAA}$ reporter suggesting that $K^+$ supports KL formation (Fig. 6d, compare blue bars of KL and KL-$TL_{GAAA}$). Increasing $Mg^{2+}$ concentrations boosted the FRET efficiency for the KL-$TL_{GAAA}$ reporter as compared to the KL reporter supporting the idea that $Mg^{2+}$ promotes $TL_{GAAA}$ docking onto the KL in vitro (Fig. 6c, right panel; d, compare blue and green bars within KL and KL-$TL_{GAAA}$).

We evaluated the ability of the KL to function as a non-canonical receptor for $TL_{GAAA}$. For this, we replaced the KL forming H22-H88 RNA sequence with an established tetraloop receptor (TLR) that is known to bind to the $TL_{GAAA}$ motif (Fig. 6c, TLR-$TL_{GAAA}$)[47,48]. Titration with increasing $Mg^{2+}$ concentrations led to a similar increase in FRET efficiency as observed for the KL-$TL_{GAAA}$ reporter (Fig. 6d, compare green bars of KL-$TL_{GAAA}$ and TLR-$TL_{GAAA}$). Moreover, both KL-$TL_{GAAA}$ and TLR-$TL_{GAAA}$ reporters exhibited similar equilibrium binding constants ($K_{eq,Mg(II)} \sim 2$ mM) for $Mg^{2+}$ (Fig. 6c, right panel, Supplementary Tables 6–8) supporting the idea that $TL_{GAAA}$ docks onto to the KL, as efficiently as a canonical TLR. We suggest that the KL functions as a receptor for H68 $TL_{GAAA}$, and the stabilization of this tertiary contact requires $Mg^{2+}$.

We monitored the stability of the KL-$TL_{GAAA}$ RNA construct by UV thermal melting in two buffers containing: (1) 100 mM $K^+$, which favors KL formation, and (2) 100 mM $K^+$ and 100 mM $Mg^{2+}$, which promotes KL formation and KL-$TL_{GAAA}$ docking. We found a single melting transition at 66 °C for the RNA construct in buffer conditions that favors KL formation, which we attributed to KL melting (Fig. 6e, upper panel, Supplementary Table 7). In the second condition, which promotes KL formation and KL-$TL_{GAAA}$ docking, the KL melting temperature increased to 77 °C (Fig. 6e, shift arrow, Supplementary Table 7). In this profile, we observed a second melting transition at 35 °C (Fig. 6e, lower panel, Supplementary Table 7). We attribute this melting transition to the undocking of the $TL_{GAAA}$ from the KL, given the significant increase in FRET efficiency observed for the KL-$TL_{GAAA}$ reporter in these buffer conditions (Fig. 6c and Supplementary Fig. 5). These data support the idea that H22 and H88 form a robust KL in the presence of monovalent $K^+$ ions. In contrast, the KL-$TL_{GAAA}$ interaction is less stable and requires the presence of $Mg^{2+}$.

Using the FRET reporters, we investigated the impact of the Puf6-Pumilio repeat domain (Puf6-PUM) on the formation of the KL and the KL-$TL_{GAAA}$ contact in vitro. We found that Puf6-PUM did not interfere with KL formation and KL-$TL_{GAAA}$ docking as judged by the increased FRET efficiency for both

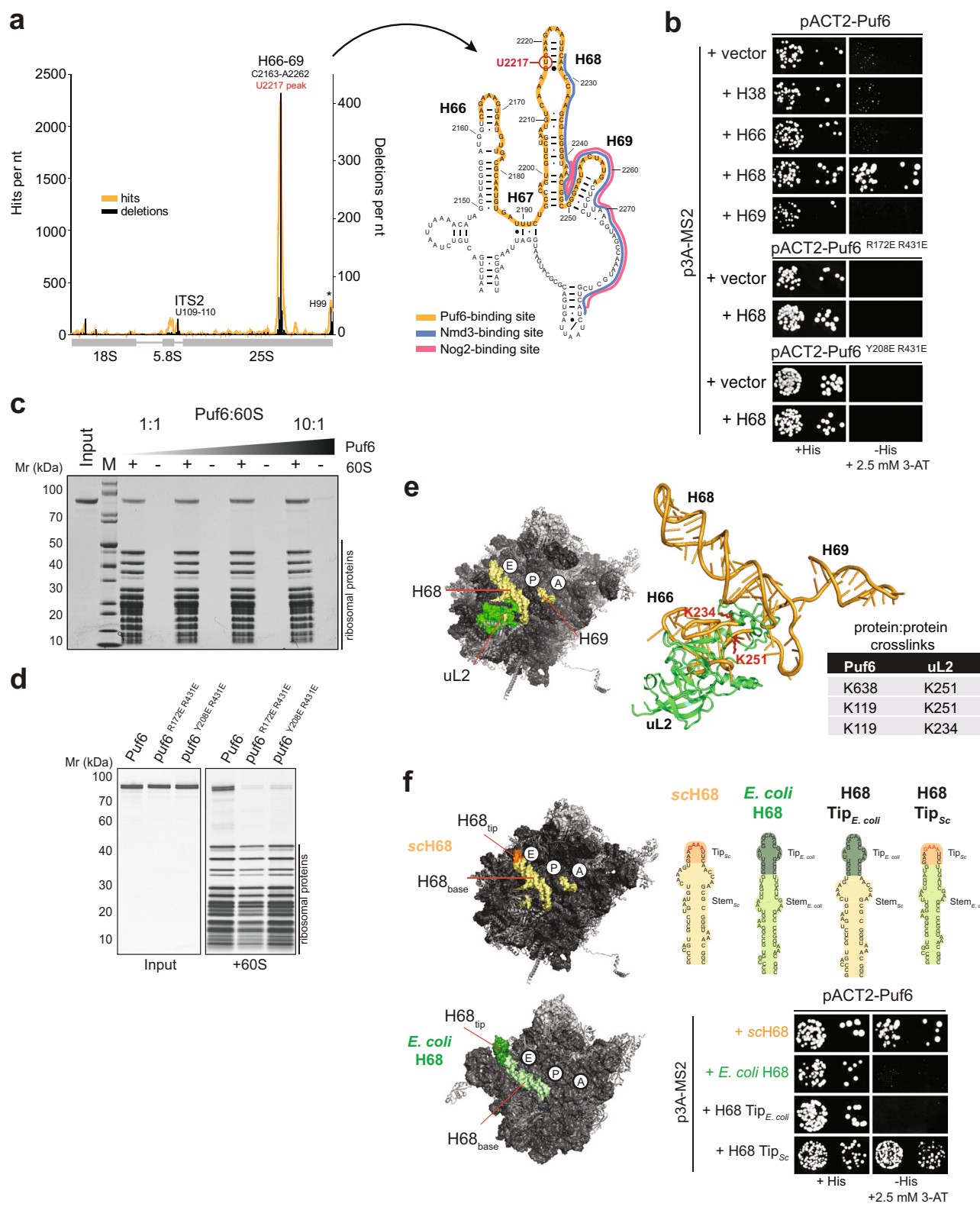

reporters with increasing $Mg^{2+}$ concentrations (Fig. 6c, right panel, red circles). Surprisingly, we observed a significant increase in FRET efficiency for the KL-$TL_{GAAA}$ reporter upon Puf6-PUM addition in buffer conditions that favor only KL formation (Fig. 6c, right panel, red circles; d, compare blue and red bars for KL-$TL_{GAAA}$). Puf6-PUM also promoted $TL_{GAAA}$ docking onto the canonical TLR in buffer conditions that do not favor TLR-$TL_{GAAA}$ interactions (Fig. 6c, right panel, red circles; d, compare

blue and red bars for TLR-$TL_{GAAA}$). Thus, Puf6 can substitute the role of $Mg^{2+}$ to promote non-canonical KL-$TL_{GAAA}$ as well as canonical TLR-$TL_{GAAA}$ tertiary contact formation (Fig. 6f) in vitro. An increase in FRET efficiency was observed for the KL reporter upon Puf6-PUM addition in the presence of KL promoting buffer (Fig. 6c, right panel, red circles; d, compare blue and red bars for KL). This very likely reflects further compaction of KL due to Puf6-mediated $TL_{GAAA}$ docking.

**Fig. 5 Puf6 binds to H68 within 25S rRNA at the subunit interface. a** CRAC analysis of Puf6 binding to the rDNA. Traces that indicate nucleotides in cDNA derived from cross-linked RNA (orange) or deleted (black) are shown. Cross-link sites are indicated. Non-specific signals for H99 (asterisk) are commonly found in CRAC experiments [38,70]. Secondary structure of H66–H68 showing Puf6 binding site (orange) and partial overlap with Nmd3 binding site (blue) and Nog2 (pink, upper panel right). **b** The indicated plasmids and p3A-MS2 vectors expressing different rRNA helices fused to MS2 RNA were transformed into yeast. Transformants were spotted in 10-fold dilution steps on selective SD-Leu-Ura (left panel) and SD-His plates containing 2.5 mM 3-AT (right panel) and grown for 3–4 days at 30 °C. **c** The 60S (150 nM) subunits were incubated with Puf6 (150 nM to 1.5 μM) for 10 min at 20 °C and layered onto a 30% (w/v) sucrose cushion. After centrifugation, the supernatant was removed, and the pellet was analyzed on a NuPAGE 4–12% Bis-Tris gradient gel. Negative controls (Input) lacking 60S subunits were treated in an identical manner to serve as a control for Puf6 protein precipitation. **d** In total, 150 nM 60S subunits were incubated with 1.5 μM puf6 mutants at 20 °C and layered onto a 30% (w/v) sucrose cushion. After centrifugation, the supernatant was removed, and the pellet was analyzed on a NuPAGE 4–12% Bis-Tris gradient gel. Negative controls (Input) lacking 60S subunits were treated in an identical manner to serve as a control for puf6 mutant protein precipitation. **e** XL-MS of a reconstituted 60S:Puf6 complex. Table summarizing the identified protein:protein cross-links showing proximity between Puf6 and uL2. uL2 (green) is in close proximity of the identified rRNA binding site of Puf6 (yellow) at the 60S subunit interface. **f** Yeast three-hybrid analysis of H68 chimeric constructs between *S. cerevisiae* and *E. coli*. pACT2-Puf6 and indicated the p3A-MS2 vectors expressing depicted chimeric rRNAs fused to MS2 RNA were transformed into yeast. Transformants were spotted in 10-fold dilution steps on selective SD-Leu-Ura (left panel) and SD-His plates containing 2.5 mM 3-AT (right panel) and grown for 3–4 days at 30 °C.

Finally, we quantitated the interaction of Puf6 with H68 alone and the minimal FRET KL-TL$_{GAAA}$ RNA by fluorescence anisotropy experiments (Fig. 6g). We found that Puf6 binds to isolated H68 albeit with lower affinity as compared to the KL-TL$_{GAAA}$ RNA in buffer conditions promoting tertiary contact formation suggesting that, like Mg$^{2+}$, Puf6 stabilizes the KL-TL$_{GAAA}$ contact.

**Puf6 primes a nucleolar 60S pre-ribosome for export.** We investigated the impact of Puf6 on the 60S assembly pathway in vivo at LT. For this, we isolated Nop7-TAP from WT and *puf6Δ* cells grown at 20 °C. Nop7 is an ideal TAP-bait to interrogate delays in 60S assembly since it is recruited to early nucleolar 60S pre-ribosomes and remains associated until late nuclear maturation stages[10]. These biochemical studies showed that Sda1 recruitment to the 60S pre-ribosome was strongly impaired in the *puf6Δ* mutant (Fig. 7a). Cryo-EM studies of 60S pre-ribosomes showed that Sda1 binds to the base of H68 that is engaged in a KL-TL$_{GAAA}$ contact (Fig. 7b). In light of these studies, we suggest that Sda1 is recruited to the 60S pre-ribosome after the KL-TL$_{GAAA}$ contact has been formed (Fig. 7b).

Sda1 forms part of a cluster of assembly factors (Rix1-Ipi-complex, AAA-ATPase Rea1, Rsa4, and the GTPase Nog2)[49], whose release from 60S pre-ribosomes licenses recruitment of the export machinery[9]. Sda1 has been proposed to promote the downstream recruitment of the Rix1-Ipi-complex and Rea1 to the 60S pre-ribosome[7,8]. Consistent with this, we observed a significant impairment in the recruitment of the Rea1 to the Nop7-TAP particle isolated from the *puf6Δ* mutant (Fig. 7a). CRAC studies suggest that the Puf6 location on the 60S pre-ribosome overlaps with the binding site of Nog2 (Fig. 4a), a placeholder for the export adaptor Nmd3[9]. Western analyses revealed that Nog2 recruitment to the Nop7-TAP particle was also severely impaired in the *puf6Δ* mutant at LT. In view of FRET studies and the biochemical data, we infer that failure to correctly form the KL-TL$_{GAAA}$ contact impairs the recruitment of checkpoint factors that license a 60S pre-ribosome for nuclear export.

## Discussion

The multitude of possible intramolecular base-pairing during early pre-rRNA folding and the increased stability of mispaired RNA intermediates at LT negatively impacts ribosome assembly[13,50]. We interrogated the yeast proteome by SWATH-MS to reveal how the ribosome assembly pathway adapts to LT. We found that yeast cells boost production of assembly factors at LT (20 °C) in comparison to optimal 30 °C, the optimal growth temperature for yeast (Fig. 1c and Supplementary Data 1). The induced factors interact with pre-ribosomes at different maturation stages and contact rRNA-rich centers including the stalk base, the ITS2 "foot" and the rRNA helices that form the catalytic subunit interface (Fig. 1c and Supplementary Figs. 2 and 3).

One LT-induced factor, Puf6, whose location on the 60S pre-ribosome remained unresolved caught our attention (Figs. 1c and 2a, b). Puf6 belongs to the Pumilio and fem3 mRNA-binding factor (PUF) family which shares an RNA binding domain composed of several α-helical repeats[25,51,52]. PUF proteins are involved in the post-transcriptional regulation of mRNA expression by binding to regulatory elements of mRNA targets[53]. Yeast Puf6 and its human ortholog Pum3/Puf-A adopt an atypical L-shaped confirmation and are the most distant member of the yeast PUF family[34,35]. In addition to the eight PUF repeats, both Pum3/Puf-A and Puf6 contain three PUF repeats at the N-terminus (Fig. 4a)[35]. Structure-guided studies demonstrated that these non-canonical repeats participate in recruiting Puf6 to the 60S pre-ribosome (Fig. 4d, e, f). The identified mutants partly overlap with the previously identified interaction surface that (Fig. 4b) contributes to Puf6 function in 7S pre-rRNA processing and ASH1 mRNA localization[35]. CRAC and yeast three-hybrid studies identified the hairpin-loop tip region within H68 of 25S rRNA as an interaction platform for Puf6. However, Puf6 does not interact with H38 and H66 that contain TL$_{GAAA}$ motifs within their hairpin-loops suggesting that additional features within the H68 stem region contribute to Puf6:H68 interactions (Fig. 5b).

On the 60S subunit, the H68 TL$_{GAAA}$ motif (rRNA domain IV) engages in an unusual tertiary contact with a kissing loop (KL) formed between H22 (rRNA domain I) and H88 (rRNA domain V) bringing together distant secondary structure elements. Cryo-EM studies suggest this contact contributes to the compaction of the flexible subunit interface (rRNA domains III–V) with the rigid solvent-exposed side (rRNA domains I, II, and VI) on the 60S pre-ribosome[8,54]. Using FRET reporters, we provide evidence for the reconstitution of the KL, and docking of the TL$_{GAAA}$ onto the KL in vitro. Our studies show that the KL functions as a non-canonical TL$_{GAAA}$ receptor, as the TL$_{GAAA}$ motif of H68 docked onto a canonical tetraloop receptor with a similar FRET efficiency. Moreover, we found that Mg$^{2+}$ is required to promote TL$_{GAAA}$ docking onto the KL similar to canonical TLR-TL$_{GAAA}$ interactions (Fig. 6d, compare green bars for KL-TL$_{GAAA}$ and TLR-TL$_{GAAA}$). Structural analyses of the 60S subunit revealed that >100 Mg$^{2+}$ ions and >50 monovalent cations are bound to rRNA, the majority of which decorate the RNA-rich subunit interface, in particular the rRNA-rich peptidyl transferase center, where stabilizing cationic tails of r-proteins are absent[55]. These

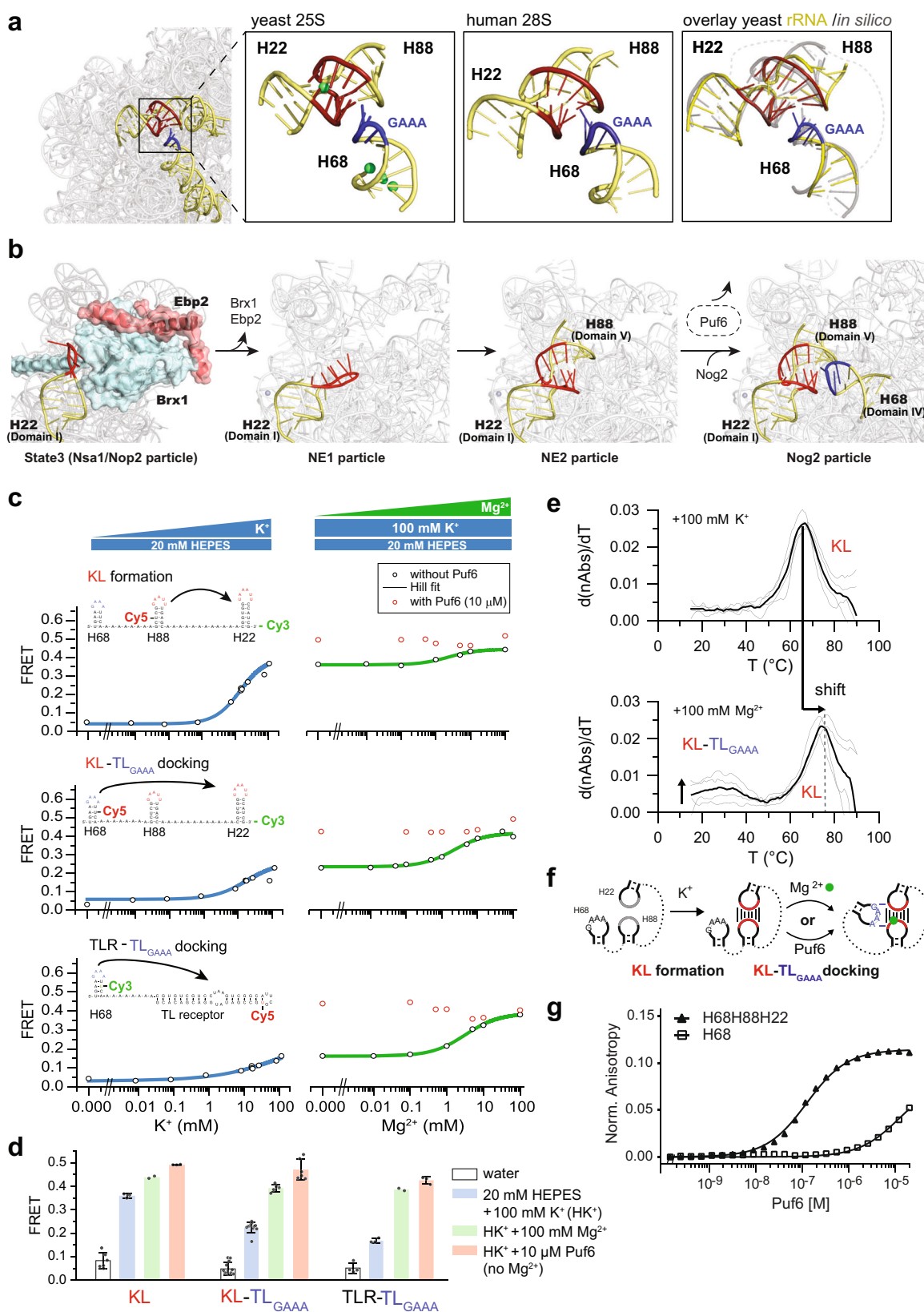

cations stabilize the tertiary structure by mediating interactions between its rRNA domains[55]. We found that Puf6 does not interfere in the formation of the KL-TL$_{GAAA}$ contact. Instead, FRET studies indicate that Puf6 replaces the need for Mg$^{2+}$ to dock the TL$_{GAAA}$ onto the KL in vitro (Fig. 6c, d, compare blue and red bar for KL-TL$_{GAAA}$). We speculate that Puf6 shields the

negative charges of the rRNA backbone to establish correct tertiary interactions between flexible rRNA helices, prior to final stabilization by Mg$^{2+}$[12,47]. Puf6 interacts only with H68, but not with isolated H22 and H88 in vivo (Supplementary Fig. 4) supporting the idea that Puf6 ushers H68 docking onto a pre-formed H22-H88 KL.

**Fig. 6 Puf6 ushers rRNA tertiary contact formation. a** Tertiary contact between H22-H88 kissing loop (KL, in red) and H68 tip (GAAA in blue) within 25S rRNA (left; PDB ID: 4V7R), $Mg^{2+}$ in green. Middle: tertiary contact between H22-H88 KL (in red) and H68 tip (GAAA in blue) within 28S rRNA (PDB ID: 4UG0) Right: knowledge-based structure prediction of the RNA construct (colored) matches well (RMSD = 1.6 Å) with the tertiary contact within 25S rRNA (gray). Dotted lines indicate poly-A linkers. **b** Events leading to tertiary contact formation (nucleolar particles, PDB ID: 6CB1 (ref. [6]); NE1 and NE2 particles, PDB ID: 6YLX and 6YLY (ref. [8]); nuclear particle, PDB ID: 3JCT (ref. [17])). Puf6 functions between NE2 and Nog2 particle, in dashed-line oval. **c** FRET efficiencies for 1 μM Cy3s (donor)/Cy5s (acceptor) labeled RNA: KL, KL-$TL_{GAAA}$, and TLR-$TL_{GAAA}$ (cartoons above plots, Supplementary Table 3) were plotted against indicated $K^+$ (left panel; 20 mM HEPES buffer) or $Mg^{2+}$ (right panel; 20 mM HEPES, 100 mM $K^+$) with (red circles) or without (black circles) 10 μM GB1-Puf6-PUM (161–656). **d** FRET increase upon addition of $K^+$ (in blue), $Mg^{2+}$ (in green), and 10 μM GB1-Puf6-PUM (in absence of $Mg^{2+}$, in red) for KL, KL-$TL_{GAAA}$ and TLR-$TL_{GAAA}$ RNA constructs. Each independent experiment is presented (dots), and error bars indicate ±SD of $n > 2$ independent experiments. **e** UV thermal melting curves for KL-$TL_{GAAA}$ in 20 mM HEPES buffer containing 100 mM $K^+$ (top) and 100 mM $Mg^{2+}$ (bottom). Horizontal arrow: KL melts at a higher temperature in presence of $Mg^{2+}$; vertical arrow: melting transition corresponding to KL-$TL_{GAAA}$ docking in presence of $Mg^{2+}$. **f** Scheme for KL-$TL_{GAAA}$ formation in presence of $K^+$, $Mg^{2+}$, and Puf6. **g** In total, 2.5 nM Cy5-labeled H68- or H68-H88-H22-RNA was titrated with 0.16 nM to 20 μM Puf6-PUM(161–656). Fluorescence anisotropy was measured using a Tecan Safire II plate reader. Puf6 binding to H68 alone and to the KL-$TL_{GAAA}$ construct was $K_D$ ~ 10 μM and $K_D$ ~ 125 nM, respectively. Source data are provided as a Source Data file.

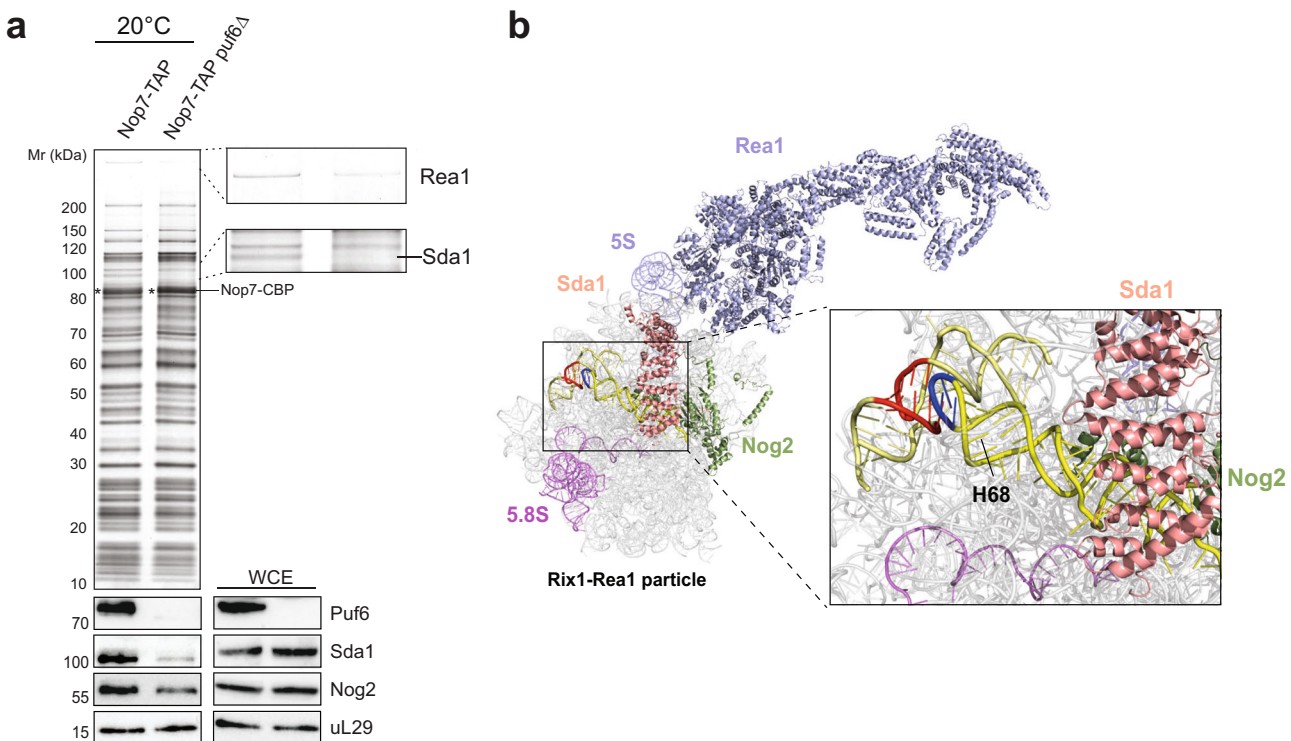

**Fig. 7 Puf6 primes a 60S pre-ribosome for nuclear export. a** Nop7-TAP particles isolated from WT and *puf6Δ* cells were grown at 20 °C and separated on a NuPAGE 4–12% Bis-Tris gradient gel, analyzed by Silver staining and Western analysis using indicated antibodies. Prior to TAP, whole cell extracts (WCE) from the indicated yeast strains were separated by SDS-PAGE and analyzed by Western blotting using indicated antibodies. uL29 was used as a loading control. The bait protein Nop7-CBP is indicated by an asterisk. **b** Puf6 binding site overlaps with Sda1 (salmon) and Nog2 (green) on the early 60S pre-ribosome (PDB ID: 6YLH[15]). Source data are provided as a Source Data file.

Cryo-EM structures of early nucleolar states implicate a non-linear pathway to compact 27S pre-rRNA. Initially, the 5′ rRNA domains I and II of 27S pre-rRNA adopt a near mature conformation through tertiary contacts with the 3′ terminal domain VI, to form the solvent-exposed backside of the 60S subunit. In contrast, the central rRNA domains III, IV, and V, which eventually form the subunit interface appear to be dynamic and are not resolved in these early states[5,6]. This open arch-like conformation of 60S pre-ribosome undergoes compaction through several long-range tertiary contacts with the central RNA domains III, IV, and V. $TL_{GAAA}$/$TL_{GNRA}$ are prevalent motifs that direct higher-order folding and compaction of RNAs. On a mature 60S subunit, there are >15 GAAA/GNRA motifs located within hairpin loops and bulges engaged in tertiary contacts[2]. A particular challenge for each of these motifs during pre-rRNA folding/compaction is to search and pair with their cognate

tetraloop receptors. A productive search is critical at LT where $TL_{GAAA}$/$TL_{GNRA}$ motifs mispaired with non-cognate receptors are stabilized, thus posing a barrier to correct rRNA folding and compaction[11–13]. We propose that yeast cells induce Puf6 production at LT to ensure that H68 $TL_{GAAA}$ establishes a correct contact with the pre-formed KL. Puf6 showed an enhanced affinity to H68 when $TL_{GAAA}$ is bound to KL then to H68 in isolation in buffer conditions promoting tertiary contact formation (Fig. 6g). These data suggest that like $Mg^{2+}$, Puf6 stabilizes the KL-$TL_{GAAA}$ tertiary contact, and the conformation of the H68 tip region within the tertiary contact is likely to be the state recognized by Puf6. Our SWATH-MS studies revealed additional non-essential assembly factors that were induced at LT (Fig. 1c). These include Nop12, Pwp1, and Arx1 whose depletion, like Puf6, renders yeast cells cold-sensitive and impairs pre-rRNA folding/processing[56,57]. Work from the Woolford laboratory revealed

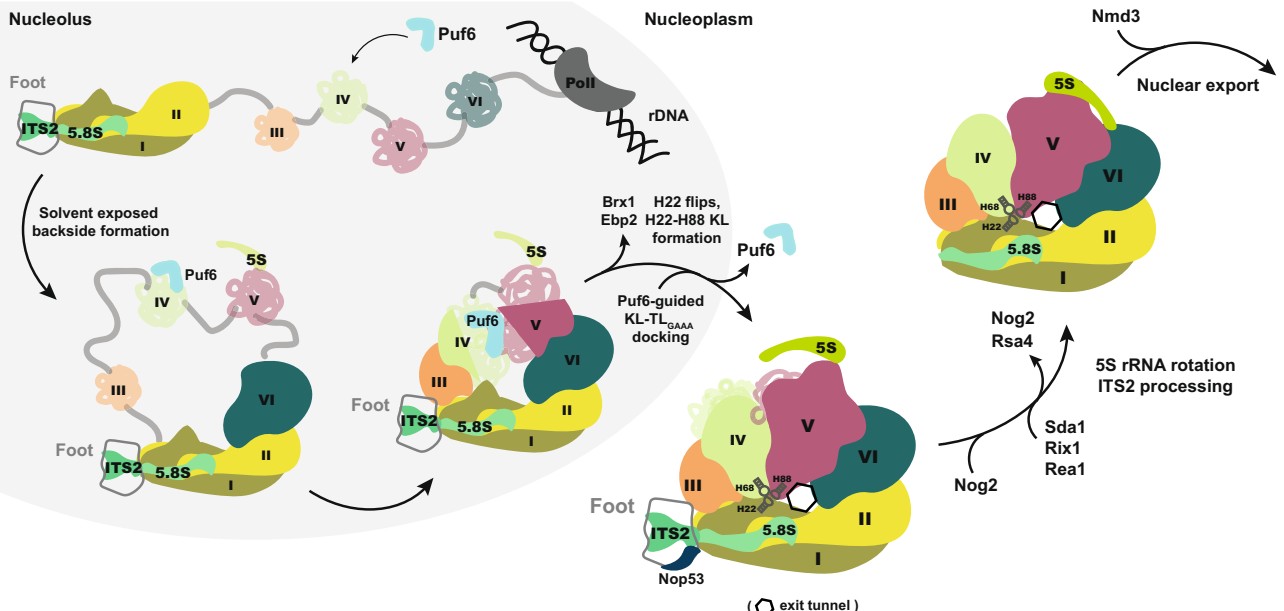

**Fig. 8 Schematic for Puf6 function during 60S assembly.** The 27S rRNA precursor co-transcriptionally associates with assembly factors/r-proteins and folds into rRNA-independent domains. The solvent-exposed side (domains I, II, and VI) establishes initial tertiary contacts which are followed by gradual compaction of domains III–V that form the catalytic subunit interface within the 60S subunit. Puf6 (light blue) is co-transcriptionally recruited to H68 within domain IV where it awaits the formation of KL between H22-H88. Release of assembly factors Brx1-Ebp2 enables a conformational change of H22 and its productive engagement with H88, forming a KL—the receptor for H68-TL$_{GAAA}$. Puf6 ushers H68 docking onto the pre-formed KL that stabilizes the subunit interface serving a binding platform for 5S RNP maturation factors (Sda1, Rix1, Rea1) and export driving machinery (Nog2, Nmd3).

how Nop12 ensures correct folding of 5.8S rRNA, thus endorsing the requirement of dedicated factors to direct pre-rRNA folding events at LT[57].

We propose a model for the order of events that lead to the formation of the KL-TL$_{GAAA}$ contact and export competent 60S pre-ribosomes (Fig. 8). The emerging 27S pre-rRNA co-transcriptionally folds into different pre-rRNA domains which recruit assembly factors and r-proteins. rRNA domains I, II, and VI fold their near-native conformation to form the solvent-exposed backside of the 60S subunit[5,6]. The transition from an open-arch to a compacted 60S pre-ribosome necessitates rRNA domains III, IV, and V to establish tertiary contacts with the folded solvent backside of the 60S subunit, including the KL-TL$_{GAAA}$ contact. While H68 and H88 are not fully resolved in early nucleolar 60S pre-ribosome intermediates, H22 (rRNA domain I) is visible and is bound to assembly factors Ebp2 and Brx1[5,6]. Ebp2:Brx1 prevents H22 to prematurely form a KL with the H88 hairpin-loop within RNA domain V (Fig. 6b). Puf6 is co-transcriptionally recruited to H68, where it awaits the formation of its cognate RNA receptor, the H22:H88 KL. Ebp2-Brx1 removal from the 60S pre-ribosome, via a yet unknown mechanism, re-orients H22 (NE1 particle) making it competent to interact with the hairpin-loop of H88 to form the KL (NE2 particle)[8]. Puf6 bound to the H68 hairpin-loop region promotes docking of the TL$_{GAAA}$ motif onto the KL and stabilizes the tertiary contact. Docking of the TL$_{GAAA}$ onto the KL is visible only on a later Nog2-particle implicating a multi-step process to connect distant H22, H68, and H88 rRNA secondary structures (Fig. 6b). Given that Puf6 and H68 are not resolved in the early NE2 state[8], and that Puf6 release precedes Nog2 recruitment to the 60S pre-ribosome (Figs. 3c, e and 6b), we suggest that Puf6-directed H68 docking onto the KL occurs between the NE2 state and Nog2 recruitment to the 60S pre-ribosome (Figs. 6b and 8).

Cryo-EM studies of the Nog2-particle suggest that H68 recruits Sda1 after it has established the KL-TL$_{GAAA}$ contact[54]. In agreement with these data, we found that, at LT, Sda1 recruitment

to 60S pre-ribosomes is severely impaired in *puf6Δ* cells (Fig. 7a). The formation of the polypeptide exit tunnel (PET) also requires correct compaction of 27S pre-rRNA. Cryo-EM studies of nuclear 60S pre-ribosomes (class R1) isolated from an *rpl4Δ63-87* mutant (which misassembles the PET) revealed a non-native H68 conformation, that also failed to recruit Sda1[58]. We speculate that correct compaction of 27S pre-rRNA licenses formation of the long-range KL-TL$_{GAAA}$ tertiary contact which in turn permits recruitment of Sda1 and checkpoint factors, that drive a correctly assembled 60S pre-ribosome towards export competence (Fig. 8).

Why are ribosome assembly factors induced at LT? A series of findings regarding misfolding of 16 rRNA within the bacterial small ribosome subunit (30S), perhaps, provide an insight[50,59,60]. These studies describe a cold-sensitive bacterial strain wherein a C23U mutation within 16S rRNA alters the base pair of the 5′ terminal pseudoknot helix to a G-U pair which presumably weakens the 5′ helix. Overexpression of a KH-domain protein RbfA, which is typically induced in wild-type *E. coli* at LT, rescued the cold-sensitive phenotype of the 16S rRNA mutant strain, suggesting that the C23U mutation decreases the affinity of RbfA for its rRNA-binding site. Notably, *E. coli* lacking the RbfA gene is cold-sensitive like the C23U mutant. Thus, it seems that the correct formation of the pseudoknot helix within 16S rRNA, at LT, requires the dedicated service of RbfA. It is tempting to speculate that, at LT, frequent pre-rRNA mispairing decreases productive encounters with the ribosome assembly machinery. Alternatively, it is possible that r-proteins and assembly factors are recruited only to correctly folded pre-rRNA, whose levels, at LT, fall significantly due to mispairing/misfolding. Therefore, increasing levels of r-proteins and assembly factors might compensate for the altered equilibrium state and steer pre-rRNA folding towards productive ribosome formation. How yeast cells sense LT to boost r-protein and assembly factor production remains unclear. Strikingly, assembly factor mutants from diverse model organisms including *E. coli*, *S. cerevisiae*, *D. melanogaster*, and *A. thaliana*, very often exhibit cold-sensitive phenotypes[49,54,58,59,61–65]. We propose that inducing

assembly factors at LT might be a conserved response to counter pre-rRNA misfolding and guarantee efficient ribosome production.

## Methods

**Yeast strains and plasmids.** The *Saccharomyces cerevisiae* strains used in this study are listed in Supplementary Table 1. Preparation of media, yeast transformations, and genetic manipulations were performed according to established procedures[66].

All plasmids used in this study are listed in Supplementary Table 2. All recombinant DNA techniques were performed according to established procedures using *Escherichia coli* XL1 blue cells for cloning and plasmid propagation. Mutations in *PUF6* were generated using the QuikChange site-directed mutagenesis kit (Agilent Technologies, Santa Clara, CA, USA). All cloned DNA fragments and mutagenized plasmids were verified by sequencing.

**SWATH-MS sample preparation for the LC-MS/MS analyses.** Three series of wildtype yeast cells (BY4741) were grown in liquid YPD media at 20 °C, 25 °C, 30 °C and 37 °C and harvested at OD$_{600}$ ~ 3. 4 OD units of the cells were collected by centrifugation at 1500 × g 5 min at 4 °C. The supernatants were discarded, and the cells were washed three times with cold (−20 °C) acetone. Cells were disrupted five times through 5 min mechanical lysis[32] at 4 °C in 8 M urea, 0.1 M NH$_4$CO$_3$, and 5 mM EDTA. The total protein amount was determined using BCA Protein Assay Kit (Thermo). In total, 1 mg of yeast proteins was reduced with 12 mM DTT at 37 °C for 30 min and alkylated with 40 mM iodoacetamide at room temperature in the dark for 30 min. Samples were diluted with 0.1 M NH$_4$HCO$_3$ to reach a final urea concentration of 1 M and digested with sequencing grade porcine trypsin (Promega, 1:100 trypsin:protein). Digestion was stopped by adding formic acid to a final concentration of 1%. Peptides were desalted using 1cc reverse-phase Sep-Pak tC18 cartridges (Waters, Milford MA) according to the following procedure: Cartridges were wetted with one volume (1 ml) 100% methanol, washed with two volumes of 80% acetonitrile, 0.1% formic acid (FA) and equilibrated with three volumes of 0.1% FA. The acidified peptides were loaded twice on the cartridge, washed with three volumes of 0.1% FA, and eluted with two volumes of 50% acetonitrile, 0.1% FA. Peptide was dried in a speedvac and resolubilized in 50 µl of 0.1% formic acid and frozen at −20 °C. The peptide concentration was measured on nanodrop and adjusted to 1 µg/µl and spiked with 1:20 (v/v) of iRT peptides[67] for mass spectrometry acquisition. 1 µg of yeast tryptic peptides was injected on a 6600 TripleTof mass spectrometer (ABSciex, Concord, Ontario) interfaced with an Eksigent NanoLC Ultra 1D Plus system (Eksigent, Dublin, CA). The peptides were separated on a 75-µm-diameter, 40 cm-long newObjective emitters packed with Magic C18 AQ 3 µm resin (Michrom BioResources) and eluted at 300 nl/min with a linear gradient of 2-to-30% Buffer A for 120 min (Buffer A: 2% acetonitrile, 0.1% formic acid; Buffer B: 98% acetonitrile, 0.1% formic acid). MS data acquisition was performed in either data-dependent acquisition (DDA, top20, with 20 s dynamic exclusion after 1 MS/MS) or data-independent acquisition (DIA) SWATH-MS mode (64 variable windows setup acquired each for 50 ms and MS1 scan for 250 ms)[68]. For either mode, the collision energy was set to 0.0625 × m/z−10.5 with a 15-eV collision energy spread regardless of the precursor charge state.

**DDA data analysis.** A detailed description for DDA search and spectral library construction[15] can be found in the Supplementary Information.

**SWATH-MS data analyses.** A detailed description of SWATH-MS data extraction[49] can be found in Supplementary Information. The nine most significant clusters as suggested by the Dmin and overlap functions of mFuzz (Supplementary Fig. 1j, k) (embedding 429 of the significantly regulated proteins) are provided in Supplementary Data 1. The dataset has been deposited to the ProteomeXchange Consortium via the PRIDE[69] partner repository with the dataset identifier PXD016320.

**Temperature shift assay and whole-cell extract preparation.** Wildtype yeast cells (BY4741) were grown until the late log phase at indicated temperatures. Cells were diluted into pre-cooled or pre-warmed media to OD600 = 0.5. Samples were retrieved at indicated timepoints and whole-cell extracts were prepared. 4 OD units were harvested by centrifugation at 5422×g for 3 min and cells were prechilled on ice for 5 min. In total, 150 µl lysis solution (1.85 M NaOH, 8% β-mercaptoethanol) was added to the cells and then lysed on ice for 10 min. In total, 150 µl 50% TCA were added and proteins were precipitated on ice for 10 min. Samples were centrifuged for 2 min at 25,199×g at 4 °C. Pellet was washed with 1 ml 100% ice-cold acetone and afterward air-dried. Pellet was resuspended in 100 µl 1× LDS sample buffer (Invitrogen, Carlsbad, CA, USA), heated for 10 min at 70 °C before loading on an SDS gel.

**CRAC analysis.** CRAC analysis was performed using the Puf6-HTP strain[38,70]. Briefly, *S. cerevisiae* cells were grown in YPD media until an OD600 of ~0.5 was reached and UV irradiated. Cells were cross-linked in the Vari-X-linker for 12 s and CRAC libraries were paired-end sequenced (50 bp) on a HiSeq2500 at Edinburgh Genomics, University of Edinburgh. The CRAC data were processed using pyCRAC-1.42 software suite (https://git.ecdf.ed.ac.uk/sgrannem/pycrac) and mapped to the reference sequence Saccharomyces_cerevisiae. R64-1-1.75 using NovoAlign-2.07 (http://www.novocraft.com). Plots of reads aligned to the 35S

rRNA sequence were made with the pyPileup module of pyCRAC. The data have been deposited in the Gene Expression Omnibus (GEO) database under accession no. GSE174587. For CRAC data analysis, data from two biological replicates were analyzed[70]; yielding essentially the same outcomes; the data set with the better coverage (Puf6-data-set II) has been used for the presented figure.

**Yeast-3-hybrid assay.** For yeast-3-hybrid assays[71] different rRNA helices were cloned into a p3AMS2-1 plasmid and Puf6 was cloned into pACT2 vector (Supplementary Table 2). Transformed L40 coat cells were spotted in 10-fold serial dilutions on selective SD-Leu-Ura and SD-His plates with indicated concentrations of 3-Amino-1,2,4-triazole (Sigma).

**Cross-linking MS.** Reconstituted Puf6:60S complexes were analyzed by XL-MS using chemical cross-linking with disuccinimidyl suberate (DSS)[72,73]. In total, 100 µM purified 60S ribosome was incubated with 200 µM Puf6 at 20 °C for 10 min in 60S binding buffer (150 mM NaCl, 20 mM HEPES pH7.5, (150 mM MgCl₂, 5 mM β-mercaptoethanol) and cross-linked with DSS-d0/d12 (25 mM stock solution in anhydrous dimethylformamide; Creative Molecules) to a final concentration of 1 mM. The reaction was incubated at 37 °C for 30 min and subsequently quenched using 50 mM ammonium bicarbonate. Cross-linked complexes were reduced, alkylated, and digested sequentially with endoproteinase Lys-C and trypsin[72,73]. The digest was fractionated with size exclusion chromatography (Superdex Peptide, GE) and fractions were analyzed by LC-MS/MS on an Orbitrap Elite mass spectrometer[72]. Data analysis was performed using xQuest/xProphet (v2.1.3) and results were filtered to a false discovery rate of <5% at the cross-linked peptide pair level[74]. The identifications are summarized in Supplementary Data 2, and the mass spectrometry data is deposited to the ProteomeXchange Consortium via the PRIDE[69] partner repository with the identifier PXD024131.

**Purification of pre-ribosomal particles.** Tandem Affinity Purification (TAP) of pre-ribosomal particles was carried out using established procedures[32]. Calmodulin eluates were separated on NuPAGE 4–12% Bis-Tris gradient gels (Invitrogen) and analyzed by Silver staining or Western blotting using indicated antibodies. Primary antibodies in the following dilutions: α-Puf6 (1:2000; this study), α-Nog2 (1:1000, this study), α-Nmd3 (1:1000, A. Johnson, University of Texas, Austin, USA), α-Sda1 (1:1000, this study), α-Nog2 (1:500, this study), α-TAP (CBP) (1:1000, Merck AG, Darmstadt, Germany #07-482), α-uL29 (1:2000)[32], α-Arc1 (1:4000, E Hurt, University of Heidelberg, Heidelberg, Germany). The secondary HRP-conjugated αrabbit antibody (Sigma-Aldrich, St Louis, MO, USA) was used at 1:2000 dilution. Protein signals were detected using immune-Star HRP chemiluminescence kit (Bio-Rad Laboratories, Hercules, CA, USA) and captured on Fuji Super RX X-ray films (Fujifilm, Japan) or digitally using ImageQuant Analyzer (GE, Healthcare).

**Fluorescence microscopy and heterokaryon assay.** Ribosome export was monitored using extrachromosomal uL18-GFP and uS5-GFP reported constructs[32]. Cells were grown at indicated temperatures to early log phase and visualized with a DM6000B microscope (Leica, Germany). Images were acquired with a fitted digital camera (ORCA-ER; Hamamatsu Photonics, Hamamatsu, SZK, Japan) using Openlab software (Perkin-Elmer, Waltham, MA, USA) and analysis was performed using ImageJ (NIH, Bethesda, Maryland, U.S) For the heterokaryon assay[32] different yeast strains were grown until OD600 of ~1. Formating, Arx1–GFP, Gar1–GFP, Puf6-GFP cells were mixed with *kar1-1* cells expressing endogenously tagged Nup82-mCherry. The mixture was concentrated onto 0.45 mm nitrocellulose filters and subsequently placed on YPD plates. After 1 h at 30 °C, filters were transferred to YPD plates containing 50 mg/ml cycloheximide and incubated for another 1–2 h before cells were analyzed by fluorescence microscopy.

**Recombinant protein expression.** Recombinant His6-Puf6 protein was expressed in *E. coli* BL21 pRARE cells (Merck Millipore, Darmstadt, Germany) at 20 °C for 18 h and 0.5 mM IPTG. His6-tagged proteins were affinity purified by gravity flow batch purification using cOmplete His-Tag purification Resin (Roche AG, Basel, Switzerland) in 500 mM NaCl, 20 mM HEPES pH 7.5, 5 mM MgCl₂, 5 mM β-mercaptoethanol, 1 mM TCEP. Eluates were concentrated and further purified through size exclusion chromatography in 150 mM NaCl, 20 mM HEPES pH7.5, 5 mM MgCl₂, 5 mM β-mercaptoethanol on a Superdex 200 10/300 GL column (GE Healthcare). For ensemble FRET, His6-GB1-Puf6-PUM (161-656aa) was expressed and purified as described above in 500 mM KCl, 20 mM HEPES pH 7.4, 1 mM DTT, 1 mM TCEP, 0.05% Triton-X. Eluates were concentrated, dialyzed into 100 mM KCl, 20 mM HEPES pH 7.5, 1 mM DTT, and further purified through size exclusion chromatography in 100 mM KCl, 20 mM HEPES pH7.5, 1 mM DTT on a Superdex 200 10/300 GL column (GE Healthcare).

**60S pelleting assay.** The binding of the purified Puf6 and its mutants to the mature 60S subunit was investigated by sucrose pelleting assay[75]. Briefly, purified proteins were incubated with 150 nM mature 60S subunit in binding buffer (150 mM NaCl, 20 mM HEPES pH 7.5, 5 mM MgCl2, 5 mM β-mercaptoethanol) for 10 min at 20 °C and layered onto a 30% (w/v) sucrose cushion. The samples were centrifuged for 105 min at 385,840 × g in a TLA-100 ultracentrifuge rotor. After

removing the supernatant, the pellet was resuspended in buffer A (500 mM NaCl, 20 mM HEPES pH 7.5, 5 mM $MgCl_2$, 5 mM β-mercaptoethanol). A negative control lacking 60S subunits was treated identically to serve as a control for protein precipitation. All samples were analyzed subsequently on a NuPAGE 4–12% Bis-Tris gradient gel (Invitrogen, Carlsbad, CA, USA).

**Ensemble FRET measurements**. Ensemble emission spectra of the fluorescently labeled (sCy3/5) RNA constructs (Supplementary Table 3) with varying amounts of mono- and divalent cations ($K^+$, $Mg^{2+}$) were measured with a fluorescence spectrometer (Cary Eclipse, Agilent) at room temperature and corrected for bleed-through and direct-excitation. FRET proximity ratios were calculated and plotted against $[K^+]$, $[Mg^{2+}]$ (with and without 10 μM Puf6-PUM). Details are provided in Supplementary Information. Data analysis and plots were made using OriginPro 2020 (64-bit) SR1 (9.7.0.188).

**RNA UV melting**. UV thermal melting curves (absorption @260 nm) of the RNA construct (Supplementary Table 7) with varying amounts of divalent cations ($Mg^{2+}$) in presences of 100 mM monovalent cations ($K^+$) were recorded with a UV/VIS absorption spectrometer (Cary 100, Agilent) between 15 and 90 °C. More details can be found in the Supplementary Information.

**Figures and structural modeling**. Figures of molecular structures were created using PyMOL (2.2.0)[76]. Sequence alignments were generated using MAFFT (v7) JalView (2.11.1.4). Structural models of Puf6 were generated with Phyre2 server[77] and visualized using PyMOL (2.2.0)[76]. Structure models of the in vitro RNA construct were generated using RNA composer[43] based on prior knowledge (kissing loop formation H22 and H88 and base-paired helices H22, H68, H88 in PDB ID: 4V7R)[2] and secondary structure prediction using ViennaRNA[78].

**Statistics and reproducibility**. Unless otherwise indicated, error bars represent the mean ± standard deviation of at least three biological replicates (i.e., $n \geq 3$). Statistical analysis was performed using Prism (version 8.4.2; GraphPad Software Inc., La Jolla, CA, USA). Experiments in Figs. 2b, d; 3a–f, 4g; 5c, d; 7a and Supplementary Fig. 4c were performed at least three times and one representative result was used for figure preparation.

**Reporting summary**. Further information on research design is available in the Nature Research Reporting Summary linked to this article.

## Data availability

The data that support this study are available from the corresponding author upon reasonable request. The datasets generated in this study are available in the following repositories: The SWATH-MS and XL-MS data analyses have been deposited to the ProteomeXchange Consortium via the PRIDE partner repository with unique identifiers PXD016320 and PXD024131, respectively. The sequencing data have been deposited in the Gene Expression Omnibus database; accession no. GSE174587. Source data are provided with this paper.

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

## Acknowledgements

We thank M. Peter and E. Hurt for generously sharing yeast strains, plasmids, and antibodies. We thank all members of the Panse laboratory for enthusiastic discussions and the Center for Microscopy and Image Analysis, UZH for maintaining the imaging equipment. M.O.-O. acknowledges support from the Boehringer Ingelheim Fonds Ph.D. fellowship program. J.J.P. was the recipient of an ETH Zurich Postdoctoral Fellowship supported by the Marie Curie Actions for People COFUND program. Research in the Granneman lab was supported by grants from the Wellcome Trust (091549 to S.G.) and a Medical Research Council Non-Clinical Senior Research Fellowship (MR/R008205/1 to S.G.). R.K.O.S. acknowledges support from the Swiss National Science Foundation and the UZH. V.G.P. is supported by grants from the Swiss National Science Foundation (310030_188527), NCCR in RNA and Disease, ETH Zurich, Novartis Foundation, Olga Mayenfisch Stiftung, and a Starting Grant Award (EURIBIO260676) from the European Research Council.

## Author contributions

Experimental design: S. Gerhardy, M.O.-O., L.G., R.B., R.V.N., A.L., E.M., J.J.P., S. Granneman, R.K.O.S., R.A., V.G.P.; experiment execution: S. Gerhardy, M.O.-O., L.G., R.B., R.V.N., A.L., E.M., J.J.P.; data analysis: S. Gerhardy, M.O.-O., L.G., R.B., R.V.N., A.L., E.M., J.J.P.; scientific input and supervision: S. Granneman, R.K.O.S., R.A., V.G.P.; writing—original draft: S. Gerhardy, M.O.-O., V.G.P.; writing—review & editing: S. Gerhardy, M.O.-O., L.G., R.B., R.V.N., A.L., E.M., J.J.P., S. Granneman, R.K.O.S., R.A., V.G.P.

## Competing interests

The authors declare no competing interests.
