## [Peer Review File · Nature Communications]

Reviewers' Comments:

Reviewer #1:

Remarks to the Author:

Gerhardy and co-workers are reporting a comprehensive investigation identifying the function of the pumilio RNA-binding protein Puf6 for yeast ribosome biogenesis at low temperatures. Overall, this is a very elegant study combining a number of complementary approaches to define the role of Puf6 on a cellular as well as molecular level. It is well written with a convincing logic and easy to follow with high-quality figures. The experiments and controls are conducted meeting all scientific standards. Having said this, my main criticism of this manuscript is regarding the overall conclusion and title: "Puf6 ushers correct ribosomal RNA compaction at low temperatures". As I will further outline below, while the presented experiments provide interesting insight into the function of Puf6, I do not agree that they fully support this title (yet). In brief, the manuscript is missing a critical experiment demonstrating the impact of Puf6 on "ribosomal RNA compaction" in the context of ribosome biogenesis.

I will first summarize what the authors convincingly demonstrate through many complementary experiments:

1. At low temperatures, several ribosome biogenesis factors including Puf6 as well as ribosomal proteins are overexpressed – presumably to aid with ribosome biogenesis under these adverse conditions which may enhance difficulties with rRNA folding. In accordance with this finding, the *puf6* deletion strain has a cold phenotype with slow growth at 20°C and 25 °C.
2. The Puf6 protein is mainly located in the nucleus and it is unable to shuttle in a heterokaryon assay suggesting that it mainly resides in the nucleus and does not or rarely relocate to the cytoplasm.
3. At low temperatures, Puf6 is directly or indirectly required for nuclear export of the large ribosomal subunit precursor and in accordance with this finding, Puf6 associates with early 60S precursors.
4. The Pumilio repeat 1N and 5 are critical for the association of Puf6 with the 60S precursor and for its nuclear export.
5. Puf6 interacts with the tip of H68 at the solvent-exposed surface of the 60S subunit and is in proximity to ribosomal protein uL2.
6. In vitro, Puf6 can mediate the docking of the tetraloop of H68 with its kissing loop receptor formed by H22 and H88 as demonstrate with a small model RNA through a series of fluorescence experiments.
7. In the absence of Puf6, the association of the ribosome biogenesis factors Rea1, Sda1 and Nog2 with the 60S precursor is reduced suggesting that Puf6 directly or indirectly facilitates the recruitment of these factors.

Based on these experimental data, I am convinced that Puf6 is a ribosome biogenesis factor that is critical for the early stages of 60S subunit formation at low temperature. Its binding site is H68, and it is direct or indirect role is to mediate the recruitment of later assembly factors which eventually leads to nuclear export. Thus, the role of Puf6 for 60S nuclear export is likely indirect.

The in vitro FRET experiments offer an intriguing hypothesis regarding the molecular mechanism of Puf6: it is conceivable that the protein indeed may usher formation of the tertiary contact between H22, H68 and H88 thereby causing a conformational change in the 60S precursor which is then allowing the binding of later ribosome biogenesis factors. However, as the in vitro system is highly reduced to a 65 nt RNA comprising three RNA helices typically far apart in the primary structure of 25S rRNA, I would regard this model at the current stage as an interesting hypothesis, but not as a proven mechanism.

To convince the reader that Puf6 indeed alters the rRNA conformation in the 60S precursor, the authors will have to assess the rRNA structure in the presence and absence of Puf6 at low temperatures. Similar to Figure 7, they could purify the Nop7-TAP complex at 20 °C in the wild-type and *puf6* deletion strain and then probe the rRNA structure through footprinting or crosslinking (CLASH). Based on the model for Puf6's mechanism, it can be predicted that the conformation/accessibility of H22, H68 and H88 should be altered.

Alternatively, the authors should change the title of the manuscript to better reflect the definite findings of this study.

In conclusion, this interesting study reveals the function of Puf6 for ribosome biogenesis which is a novel and highly interesting finding. Critically, the manuscript also reveals a novel layer of supporting ribosome synthesis at low temperature through a number of factors of which Puf6 is characterized as a prototype. Together, these insights will receive strong attention in the entire field of ribosome synthesis as well as the area of RNA-binding proteins. The manuscript will inspire a broader look at ribosome synthesis, its regulation and mechanism at adverse conditions far beyond the classically studied optimal growth conditions which are common in the lab, but not realistic in terms of evolution.

Minor suggestion:

- line 66: delete "with"
- Is it possible to distinguish whether Puf6 is predominantly located in the nucleolus or whether it is also found in the nucleus?
- When describing amino acid substitutions in the Puf6 protein, the term "mutation" should be avoided as it refers to DNA.
- The choice of amino acid substitutions in Puf6 should be better justified. Why did the authors change all residues to glutamate? Why did they decide to substitute Y208 although the sequence alignment indicates that it is not conserved? Why did they only create double-substitutions in two pumilio repeats rather than the single substitutions?
- Line 303: What is meant with "promiscuous interactions with non-cognate tetraloop receptors"? As the authors are not investigating the interaction of Puf6 with RNAs other than H68 and H22-H88, this sentence must be rephrased.
- Line 447ff: the authors propose that "Puf6 replaces the role of Mg²⁺ to usher TL(GAAA) to dock onto the KL". This raises the question whether Puf6 is critical for ribosome biogenesis not only at low temperatures, but also under other stress conditions such as low salt. Have the authors screened the puf6 deletion strain for phenotypes under other conditions?
- line 575ff: in several cases "micro" is abbreviated as "u" instead of the Greek letter.

Ute Kothe

Reviewer #2:

Remarks to the Author:

The authors of this manuscript, "Puf6 ushers correct ribosomal RNA compaction at low temperature", present a proposed mechanism for proper pre-ribosomal RNA folding at low temperatures in yeast by the protein Puf6, a nucleolar 60S assembly factor. The authors use multiple approaches to explore the interaction of Puf6 with rRNA and with other proteins in order to reach their conclusion, which overall seems reasonable and interesting. However, several lines of inquiry seem to be either poorly supported or not well clarified, such as the claim that Mg²⁺ is required for the rRNA to fold properly and that Puf6 replaces the role of Mg²⁺ based on a FRET study.

I recommend publication of the manuscript with the following major and minor revisions to support the interpretation and the presentation of the manuscript's main points.

Major issues:

(1) The proteomics as presented in Figure 1 to set up the experimental hypothesis seems to distract from the main message and does not inherently support the authors' conclusion that "yeast cells globally boost production of ribosome assembly factors and r-proteins at [low temperature]". This may be due to unclear logic presented in the text or perhaps the data is too noisy to truly reach the same conclusion without a bit of a reach in the bioinformatic data massaging. The following suggestions would make it easier to reach the same conclusion that the authors reach regarding ribosome assembly factors and r-protein abundance:

(1a) The authors present mass spectrometry proteomics data in Figures 1a and 1b which use different levels of quantification -- in Figure 1a, peptides are hierarchically clustered while in Figure 1b proteins are profiled. These figures should use the same level of measurement, and so Figure 1a should cluster proteins not peptides.

(1b) I applaud the authors for the experimental detail included in Supplemental Table 1, this is extremely helpful for reproducibility. Figure 1b presents 9 clusters -- how was the number of 9 clusters reached? In addition, how many proteins are present in multiple clusters? From Supplemental Table 1, there's only about 50 proteins per cluster, which seems very low to be performing a functional annotation enrichment. An UpSet plot would quickly visualize the size of these clusters and the overlap of proteins between clusters, or some sort of PCA or other dimensionality reduction to show that there are indeed 9 clusters would help bolster confidence in this result.

(1c) The logic from lines 104-143 is unclear, perhaps due to confusion from figures 1a and 1b. The main message seems to be that "yeast cells globally boost production of ribosome assembly factors and r-proteins at LT" -- if you perform a post-hoc test between the low temperature and control/high temperature proteomes, is that what you see?

(2) The FRET experiments are interesting, but seem to reach a conclusion not fully supported by the data, namely that Puf6 replaces Mg²⁺ in assisting rRNA folding (lines 364-366). Perhaps the text is just not explicitly clear here, but it currently reads as though the authors are suggesting that this is an *in vivo* mechanism. Were these experiments performed in cells or *in vitro*, or cell lysate in a tube? What impact would low temperature have on magnesium levels in cells, and therefore require replacement of Mg²⁺ with Puf6 for proper rRNA folding? This section seems promising but I couldn't quite follow the logic within the context of the larger message.

Minor points:

- Section "Puf6 removal from the 60S pre-ribosome precedes Nog2 recruitment" would benefit from a very brief reminder of why these particular proteins (Ssf1, Rix1, Arx1, Kre35) were chosen as baits for affinity purification in previous section "Puf6 co-enriches with early nucleolar 60S pre-ribosomes".

- Section on FRET experiments ("Puf6 promotes an unusual tertiary contact within the 60S pre-ribosome") may benefit from clarification or rewording when explicitly describing the known role of K⁺ and Mg²⁺ in proper RNA folding (lines 314-316) to establish why these cations were chosen for the FRET studies.

Reviewer #3:

Remarks to the Author:

This manuscript identifies proteomic changes in yeast grown at different temperatures. The authors find that the expression levels of many ribosome assembly factors and ribosomal proteins increase under low temperature growth conditions. The classification of groups of proteins is quite interesting. The authors focus deeply on the function of one protein, Puf6, whose expression levels are high when yeast are grown at low temperature. The authors build a compelling story demonstrating that Puf6 binds to H68 in the 25S rRNA and assists its interaction with a kissing loop formed between H88 and H22. The combination of *in vivo* and *in vitro* experiments is well presented and thorough. I have only a few minor comments that can be addressed by editing the manuscript. I also suggest a couple of seemingly simple experiments that might make the story more complete, but they are not necessary to solidify conclusions. I support publication of this manuscript in Nature Communications, believing it will be of broad interest and impactful.

Suggested experiments:

Test single point mutants R172E and Y208E for function.

Test Puf6 point mutants and human PUM3 in RNA-binding assays (yeast 3-hybrid and/or FP).

Lines 155-156, 212, 421-424: The description of the Pumilio repeat domain of Puf6 is not quite accurate and should cite reference #48 and Zhang, et al. 2012 (<https://link.springer.com/article/10.1007/s11033-011-0785-3>) that show the divergent Pumilio repeats in this class of PUF proteins. Other than structural similarity, the Pumilio repeat domain in Puf6 is not the same as the well-characterized classical PUF proteins (yeast Puf3/Puf4/Puf5). As a result, Fig. 4a is a little misleading.

It is important to note that the RNA recognition properties of this domain in Puf6 are not equivalent to that of the eight-repeat single-stranded RNA-binding domains. Many of the Pumilio repeats in Puf6 lack the typical RNA base recognition side chains, especially in the C-terminal eight repeats that appear similar in shape to the single-stranded RNA-binding PUF proteins. These C-terminal eight repeats should not be referred to as canonical PUF repeats. Surprisingly, the N-terminal repeat (N-R1) is most similar to repeats found in Pumilio or yeast Puf3/Puf4/Puf5. The study here presents some important new information that could shed light on the mysterious nucleic acid recognition by Puf6. The authors could discuss this. Do they think Puf6 recognizes the sequence or structure of H68?

Lines 215-216: This section should include a reference to the small-angle X-ray scattering data in reference #48 that indicated a similar fold for yeast Puf6 compared to the crystal structure of human PUM3.

Lines 216-220: It would be helpful to include the equivalent residues in the human PUM3 protein in the figure legend. The authors should also point out that the mutated residues are in the RNA-binding motifs of Pumilio repeats and are near a DNA molecule that was crystallized with human PUM3. R172 was part of a cluster of residues that were mutated previously and shown to be important for growth rate, nucleic acid interaction *in vitro* as well as 7S pre-rRNA processing and ASH1 localization *in vivo*. Y208 is also part of the same patch, and K207 was mutated. R431 was mutated as part of a patch that showed no defects. It suggests that the single mutations might be sufficient to disrupt Puf6 activity. Did the authors test the single mutations? These results should be discussed relative to the published mutation patches as they further define critical residues.

Lines 247-248, 304-308: Please include specific information about what nucleotides were included in the RNAs tested in the yeast 3-hybrid assays. This is particularly important because the authors report binding to H68 by yeast 3-hybrid assay (Fig. 5b and f), but binding to H68 was weak by FP (Fig. 6g). The sequence is provided for the FRET and FP assays, which indicates a minimal model RNA was created. This should be fully described in the text and an explanation of its validation is needed. In Figure 6a (right panel), it would be helpful to indicate the polyA linkers by dashed lines to see where the connections are. In addition, it does not seem quite right to call it a *de novo* model, because it uses secondary structure information in the prediction. It would be clearer to include a brief description of the method in the figure legend. The term "knowledge-based structure prediction" in Suppl. Table 5 is a good description. Please present the data in Fig. 6g in the Results, not the Discussion.

Lines 257-261: It might be helpful to list the specific crosslinks in the manuscript text or to add a title above the table, Protein:Protein Crosslinks. The authors should also discuss the resolution of the crosslink data (how close residues need to be for crosslinking). It appears that K234 and K251 in uL2 are relatively close together, but K119 and K168 in Puf6 are near the N- and C-termini of the Pumilio repeat domain and quite far apart. A reference to Suppl. Table 2 is needed.

Lines 1153-1154: It is probably better to call it the Pumilio repeat domain rather than the Pumilio domain.

Lines 1157-1158: Isn't the reason that residues 118-654 are reliably modeled because the equivalent region of human PUM3 is in the crystal structure?

Lines 1198-1201: This summary of the conclusion is confusing to me. Do you mean that the combination of the protein:protein crosslinks and RNA-binding data indicate that Puf6 binds to H68 and close to the uL2 binding site?

Figure 5a: What are the asterisks? Could you indicate the nucleotides at the beginning and end of the major peak in the CRAC data and RNA drawing?

Figure 6c: The placement of the graph legend in the top right panel is crowded and it confused me. Could it be made smaller and/or moved above the graph?

For full transparency, the full gel blots should be included as supplementary information for figures that show fragments of Western blots.

“Puf6 primes 60S pre-ribosome nuclear export at low temperature”

(NCOMMS-21-11977-T)

Point-to-Point response (in blue)

REVIEWER COMMENTS (in black)

Reviewer #1 (Remarks to the Author):

Gerhardy and co-workers are reporting a comprehensive investigation identifying the function of the pumilio RNA-binding protein Puf6 for yeast ribosome biogenesis at low temperatures. Overall, this is a very elegant study combining a number of complementary approaches to define the role of Puf6 on a cellular as well as molecular level. It is well written with a convincing logic and easy to follow with high-quality figures. The experiments and controls are conducted meeting all scientific standards. Having said this, my main criticism of this manuscript is regarding the overall conclusion and title: “Puf6 ushers correct ribosomal RNA compaction at low temperatures”. As I will further outline below, while the presented experiments provide interesting insight into the function of Puf6, I do not agree that they fully support this title (yet). In brief, the manuscript is missing a critical experiment demonstrating the impact of Puf6 on “ribosomal RNA compaction” in the context of ribosome biogenesis.

I will first summarize what the authors convincingly demonstrate through many complementary experiments:

1. At low temperatures, several ribosome biogenesis factors including Puf6 as well as ribosomal proteins are overexpressed – presumably to aid with ribosome biogenesis under these adverse conditions which may enhance difficulties with rRNA folding. In accordance with this finding, the *puf6* deletion strain has a cold phenotype with slow growth at 20°C and 25 °C.

2. The Puf6 protein is mainly located in the nucleus and it is unable to shuttle in a heterokaryon assay suggesting that it mainly resides in the nucleus and does not or rarely relocate to the cytoplasm.

3. At low temperatures, Puf6 is directly or indirectly required for nuclear export of the large ribosomal subunit precursor and in accordance with this finding, Puf6 associates with early 60S precursors.

4. The Pumilio repeat 1N and 5 are critical for the association of Puf6 with the 60S precursor and for its nuclear export.

5. Puf6 interacts with the tip of H68 at the solvent-exposed surface of the 60S subunit and is in proximity to ribosomal protein uL2.

6. In vitro, Puf6 can mediate the docking of the tetraloop of H68 with its kissing loop receptor formed by H22 and H88 as demonstrate with a small model RNA through a series of fluorescence experiments.

7. In the absence of Puf6, the association of the ribosome biogenesis factors Rea1,

Sda1 and Nog2 with the 60S precursor is reduced suggesting that Puf6 directly or indirectly facilitates the recruitment of these factors.

Based on these experimental data, I am convinced that Puf6 is a ribosome biogenesis factor that is critical for the early stages of 60S subunit formation at low temperature. Its binding site is H68, and its direct or indirect role is to mediate the recruitment of later assembly factors which eventually leads to nuclear export. Thus, the role of Puf6 for 60S nuclear export is likely indirect.

The *in vitro* FRET experiments offer an intriguing hypothesis regarding the molecular mechanism of Puf6: it is conceivable that the protein indeed may usher formation of the tertiary contact between H22, H68 and H88 thereby causing a conformational change in the 60S precursor which is then allowing the binding of later ribosome biogenesis factors. However, as the *in vitro* system is highly reduced to a 65 nt RNA comprising three RNA helices typically far apart in the primary structure of 25S rRNA, I would regard this model at the current stage as an interesting hypothesis, but not as a proven mechanism. To convince the reader that Puf6 indeed alters the rRNA conformation in the 60S precursor, the authors will have to assess the rRNA structure in the presence and absence of Puf6 at low temperatures. Similar to Figure 7, they could purify the Nop7-TAP complex at 20 °C in the wild-type and *puf6* deletion strain and then probe the rRNA structure through footprinting or crosslinking (CLASH). Based on the model for Puf6's mechanism, it can be predicted that the conformation/accessibility of H22, H68 and H88 should be altered. **Alternatively, the authors should change the title of the manuscript to better reflect the definite findings of this study.**

We agree with the Reviewer that our data provide a basis for proving Puf6 ushers the formation of a specific tertiary contact between a H22-H88 kissing loop (KL) and H68 GAAA-tetraloop (TL) *in vivo*. However, we believe different *in vitro* and *in vivo* findings presented in this study support our inference regarding the proposed Puf6 function.

CRAC analyses and yeast three-hybrid studies show that Puf6 specifically binds to the tip region of H68 rRNA. In an early nucleolar 60S pre-ribosome, H22 rRNA does not engage with H88 rRNA to form the KL, as the binding site of H22 is occluded by Ebp2 and Brx1 (Fig. 6b, State3 Nsa1/Nop2 particle). We therefore wondered whether Puf6 binds to H68 and prevents it from forming the tertiary contact with the H22:H88 KL. In order to test this idea, we developed a 65 nucleotide RNA system that recapitulates the tertiary contact between H68 TL and H22:H88 KL *in vitro*. Cryo-EM studies (Sanghai et al., 2018; Kater et al., 2020; Wu et al., 2016; Wilson et al., 2020) indicate that KL-TL interactions are developed only on a compacted state of the 60S pre-ribosome (Fig. 6b, NE2 particle and Nog2 particle), and are absent on an open extended arch-like structure seen in State 3 Nsa1/Nop2 particle. Therefore, this minimal RNA system has merit to test our hypothesis.

Using the labeled RNA constructs and FRET studies, we show that Puf6 does not interfere in the formation of the KL-TL contact *in vitro*. Instead, the data indicate that Puf6 substitutes the role of Mg²⁺ ions to usher and stabilize the KL-TL tertiary contact *in vitro* (Fig. 6d, KL-TL_{GAAA}). Cryo-EM studies from Woolford laboratory (Fig. 4c in Wilson et al., 2020 Nat. Comm., see below) show that Sda1 recruitment to the base of H68 on the 60S pre-ribosome requires docking of H68 onto the KL. In that study, alterations in 25S rRNA structure due to an unassembled polypeptide exit tunnel

impaired H68 to pair with the KL. Consequently, the position of H68 rRNA helix clashed with the Sda1 binding site (Figure 4c in Wilson et al., 2020 Nat. Comm.; see below), thus preventing the recruitment of downstream nuclear export check point factors. In our case, at low temperature (LT), 60S pre-ribosomes isolated from *puf6* Δ cells are strongly impaired in recruiting Sda1 and downstream checkpoint factors leading to the inference that Puf6 ushers H68 to form a tertiary contact with H22:H88 KL in vivo.

Figure 4c taken from Wilson et al., 2020 NatComm

Nop7 joins the 60S pre-ribosome during early biogenesis steps and is released from the particle prior to nuclear export. This makes Nop7-TAP is a powerful bait to monitor delays in 60S pre-ribosome assembly at the protein level. However, we are hesitant to probe the H68 rRNA structure by footprinting or crosslinking analyses using the Nop7-TAP particle from WT and *puf6* Δ cells for the following reason: Based on cryo-EM studies, we anticipate H68 rRNA helix to flip over if it does not pair with the KL i. e. only the orientation of the H68 changes when it engages with the KL (Figure 4c in Wilson et al. 2020 Nat. Comm., see above), thus preventing Sda1 recruitment. This change, in our opinion, would be small to probe through footprinting or crosslinking given the conformational heterogeneity of pre-RNA within the Nop7-TAP particle.

A definitive proof requires a high-resolution structure of Puf6 bound to the tertiary contact on the 60S pre-ribosome. Therefore, we have opted, as suggested by the Reviewer, to change the title of the manuscript to “Puf6 primes 60S pre-ribosome nuclear export at low temperature” that reflects our findings.

In conclusion, this interesting study reveals the function of Puf6 for ribosome biogenesis which is a novel and highly interesting finding. Critically, the manuscript also reveals a novel layer of supporting ribosome synthesis at low temperature through a number of factors of which Puf6 is characterized as a prototype. Together, these insights will receive strong attention in the entire field of ribosome synthesis as well as the area of RNA-binding proteins. The manuscript will inspire a broader look at ribosome synthesis, its regulation and mechanism at adverse conditions far beyond the classically studied optimal growth conditions which are common in the lab, but not realistic in terms of evolution.

We thank this Reviewer for her constructive suggestions to improve our manuscript.

Minor suggestion:

- line 66: delete “with”

Corrected (page 3, line 62-65)

- Is it possible to distinguish whether Puf6 is predominantly located in the nucleolus or whether it is also found in the nucleus?

Based on colocalization with an established nucleolar marker Gar1-mCherry (Fig. 3a and 4g) we concluded that the Puf6-GFP is predominantly located in the nucleolus (page 7, lines 176-178). These findings are consistent with biochemical purifications which show that Puf6 co-enriches with the nucleolar Ssf1-TAP particle (Fig. 3c). We

do find lower amounts of Puf6 co-enriching with the nucleoplasmic Rix1-TAP (Fig. 3c). It appears that Puf6 travels with the 60S pre-ribosome from the nucleolus to the nucleoplasm, where it is released and recycled back, after performing its duty.

- When describing amino acid substitutions in the Puf6 protein, the term “mutation” should be avoided as it refers to DNA.

This has been corrected.

- The choice of amino acid substitutions in Puf6 should be better justified. Why did the authors change all residues to glutamate? Why did they decide to substitute Y208 although the sequence alignment indicates that it is not conserved? Why did they only create double-substitutions in two pumilio repeats rather than the single substitutions? Previously, based on a Puf-A:dsDNA structure -Qui et al 2014- identified a Puf6 basic patch mutant (5 basic residues were substituted to Ala) that was impaired in 7S pre-rRNA processing and ASH1 mRNA localization.

The choice of our substitutions is better explained in the text (page 8, line 212-219). We changed several residues in Puf6 to glutamic acid (Supplementary Fig. 4b,c,d) with the aim of impairing interaction with the negatively charged phosphate backbone of the RNA substrate. We found that single substitutions were not growth impaired at 20°C (Supplementary Fig. 4b). Notably, Puf6-R172,431E and Puf6-Y208,431E, but not other double mutants were growth impaired at 20°C. Hence, these double mutants were used for functional studies (Fig. 4d, 4e). Our substitutions partly map to the Puf6 surface which binds to the dsDNA substrate (Supplementary Fig. 4d). We now include growth analyses of the R172E and Y208E single mutants (red), as well as three further mutants (blue) in Supplementary Fig. 4b,c,d.

- Line 303: What is meant with “promiscuous interactions with non-cognate tetraloop receptors”? As the authors are not investigating the interaction of Puf6 with RNAs other than H68 and H22-H88, this sentence must be rephrased.

Perhaps the context of our statement was not clear.

In early nucleolar states, H22 within rRNA domain I is bound to assembly factors Ebp2 and Brx1 presumably to prevent premature KL formation (Fig. 6b, State 3, left panel). The rRNA domains IV and V that contain H68 and H88, respectively, are not fully resolved in these states (Fig. 6b). These observations led us to ask the question whether Puf6 hinder non-productive interactions of H68 with other tetraloop receptors present within pre-rRNAs.

We have rephrased it to make this point clearer for the reader. The sentence (page 10-11, line 307-311) now reads: “We wondered whether Puf6 binding prevents H68-TL_{GAAA} to undergo non-productive interactions with other, non-cognate tetraloop receptors within the pre-rRNA.”

- Line 447ff: the authors propose that “Puf6 replaces the role of Mg²⁺ to usher TL(GAAA) to dock onto the KL”. This raises the question whether Puf6 is critical for ribosome biogenesis not only at low temperatures, but also under other stress conditions such as low salt. Have the authors screened the puf6 deletion strain for phenotypes under other conditions?

We apologize to the Reviewer for not clearly stating that: “Puf6 replaces the role of Mg^{2+} in tertiary contact formation”, is based on in vitro FRET studies. We now explicitly state this (page 15, line 453-455). rRNA folding in vitro is sensitive to Mg^{2+} and K^+ cations. While K^+ shields the negatively charged phosphate backbone to facilitates secondary structure element formation (such as RNA helices), Mg^{2+} contributes to the process of binding and/or docking of distant RNA structure elements. Specifically, Mg^{2+} alleviates the electrostatic stress arising from closely packed phosphates to about the same extent as two K^+ ions, but at a lower entropic cost because fewer ions are confined near the RNA. RNA tertiary contacts both in vitro and in vivo are stabilized by Mg^{2+} . We found that Mg^{2+} -mediated docking of the TL to the KL can be replaced by Puf6 in vitro. This led us to suggest that Puf6 replaces the role of Mg^{2+} in stabilizing the KL-TL tertiary contact in vitro. We speculate that during pre-rRNA compaction in vivo Puf6 mimics the role of Mg^{2+} by ushering a stable tertiary contact and is replaced by Mg^{2+} at later assembly steps.

High-throughput genetic screens (<https://www.yeastgenome.org/locus/S000002904>), revealed that the *puf6* Δ mutant show a decreased stress resistance to low zinc concentrations (1 μ M zinc chloride, North et al 2012 PLOS Genetics) and osmotic stress (1M NaCl, Yoshikawa et al 2009 FEMS), perhaps indicative of a role of a Puf6 beyond ribosome assembly at cold temperatures. We had screened the *puf6* Δ mutant over 30 different conditions including temperatures, osmotic stress, high salt conditions, different types of starvation, drug treatments as described in Vieitez et al., 2020, BioRxiv. We found that the growth of the *puf6* Δ mutant was only sensitive to cold (20°C), but not in other conditions. We had investigated whether Puf6 protein expression is induced when WT yeast cells were subject to a hypotonic stress including shift to water and 1.2 M sorbitol. However, we do not observe any striking induction of Puf6 protein levels in these conditions (see below).

Legend: Whole cell extracts (WCE) from WT yeast cells grown at 37°C in YPD media were shifted to 20°C, water or 1.2M Sorbitol were separated by SDS-PAGE and analyzed by Western blotting using a Puf6-specific antibody. Gsp1 was used as a loading control.

- line 575ff: in several cases “micro” is abbreviated as “u” instead of the Greek letter. This has been corrected.

Reviewer #2 (Remarks to the Author):

The authors of this manuscript, “Puf6 ushers correct ribosomal RNA compaction at low temperature”, present a proposed mechanism for proper pre-ribosomal RNA folding at low temperatures in yeast by the protein Puf6, a nucleolar 60S assembly factor. The authors use multiple approaches to explore the interaction of Puf6 with rRNA and with other proteins in order to reach their conclusion, which overall seems reasonable and interesting. However, several lines of inquiry seem to be either poorly supported or not well clarified, such as the claim that Mg²⁺ is required for the rRNA to fold properly and that Puf6 replaces the role of Mg²⁺ based on a FRET study. I recommend publication of the manuscript with the following major and minor revisions to support the interpretation and the presentation of the manuscript's main points.

Major issues:

(1) The proteomics as presented in Figure 1 to set up the experimental hypothesis seems to distract from the main message and does not inherently support the authors' conclusion that “yeast cells globally boost production of ribosome assembly factors and r-proteins at [low temperature]”. This may be due to unclear logic presented in the text or perhaps the data is too noisy to truly reach the same conclusion without a bit of a reach in the bioinformatic data massaging. The following suggestions would make it easier to reach the same conclusion that the authors reach regarding ribosome assembly factors and r-protein abundance:

(1a) The authors present mass spectrometry proteomics data in Figures 1a and 1b which use different levels of quantification -- in Figure 1a, peptides are hierarchically clustered while in Figure 1b proteins are profiled. **These figures should use the same level of measurement, and so Figure 1a should cluster proteins not peptides.**

We thank the Reviewer for pointing this mistake. All cluster analysis shown have been performed on protein profiles and this is stated in Material and Methods. Fig. 1a is a schematic to illustrate the overall workflow and analysis. The experimental protein profiles of this study are shown in Fig. 1b. We have now updated the text (page 4, lines 100-104) and amended the Figure Legend accordingly (page 31, line 990-991): “hierarchical clustering of proteins based on their abundance profile at different temperatures was performed.”

(1b) I applaud the authors for the experimental detail included in Supplemental Table 1, this is extremely helpful for reproducibility. Figure 1b presents 9 clusters -- **how was the number of 9 clusters reached?**

This reviewer is correct to state that the determination of the optimal number of clusters is a challenging task, especially for short sample series and overlapping clusters. For estimating the optimal number of clusters, we relied on the internal function “Dmin” supplied with the package mFuzz. As per mFuzz's vignette, Dmin computes the “average minimum centroid distance between two cluster centers produced by the c-means clusterings for a given range of cluster number”. In practice, the optimal cluster number is determined by a significant ‘drop’ of minimum centroid distance plotted versus a range of cluster number followed by a slower decrease of the minimum centroid distance for higher cluster number (Schwaemmle and Jensen Bioinformatics, Vol. 26 (22), 2841-2848, 2010). In the Dmin plot generated, we observe a stalled decrease in the minimal centroid distance between clusters between 8-10; we therefore decided to proceed with 9 as the number of clusters for the analysis.

As a second diagnostic, we relied on the internal mFuzz function “overlap” to compute the overlap of the clusters produced by mFuzz which returns the corresponding plot based on a principal component analysis of the cluster centers (Futschik and Charlisle, *Journal of Bioinformatics and Computational Biology*, 3 (4), 965-988, 2005). In essence, this plot provides a visual representation of the proximity/similarity of the different clusters. In that PCA plot, we observe that clusters 8 and 6 are indeed very close/similar, as can also be judged by the protein profiles that are used in the manuscript. However, we considered that the step-wise increase in protein abundance in those two clusters between the temperatures 20°C and 25°C might be of biological interest and we decided indeed to keep those 9 clusters instead of the 8 clusters that would have indeed been the minimal number of clusters for maximizing the distance. As requested by this reviewer, those two diagnostic mFuzz plots (“Dmin” and PCA/“overlap”) are now provided and discussed in Supplementary Fig. 1 as supporting evidence for the choice in the number of clusters used in that study.

In addition, how many proteins are present in multiple clusters? From Supplemental Table 1, there’s only about 50 proteins per cluster, which seems very low to be performing a functional annotation enrichment.

Regarding the number of proteins per clusters, this reviewer is correct: those range from 26 (cluster 4) and 80 (cluster 7) as is reported in the Suppl Table 1. Though the number of proteins is low for each list, it seems that the DAVID analysis is sensitive enough for lists of that size to get a significant enrichment. To confirm this, we performed a DAVID analysis of the protein clustered in clusters 1-2-3 combined together to reach now 120 proteins. Those are essentially all the proteins showing a down-regulated abundance profile between 20°C and 37°C. Upon DAVID analysis we confirmed that the top 2 enriched GO-terms were again rRNA processing and ribosome biogenesis, confirming the results of the original clusters 1-2-3. Note that an independent GSEA analysis with webgestalt.org confirmed as well the enrichment for the ribosomal proteins for the down-regulated proteins between 20°C and 37°C (see point 1c below).

An UpSet plot would quickly visualize the size of these clusters and the overlap of proteins between clusters, or some sort of PCA or other dimensionality reduction to show that there are indeed 9 clusters would help bolster confidence in this result.

Unfortunately, we did not clearly state that the protein clusters generated by mFuzz are mutually exclusive (i. e. a protein profile and therefore a protein name entry can only appear in one and only one cluster). The UpSet plot suggested by this reviewer

could therefore not be performed because there is no overlap of proteins between the clusters. This is now clarified in the manuscript text (page 4, lines 103-104): “426 proteins were organized into 9 mutually exclusive clusters (Fig. 1b)”. The suggested PCA plot representing the distance between the clusters was generated using the mFuzz overlap function (see above) and is provided in the Supplementary Figure 1 to confirm the rational of the 9 clusters decided upon in that analysis.

(1c) The logic from lines 104-143 is unclear, perhaps due to confusion from figures 1a and 1b. The main message seems to be that “yeast cells globally boost production of ribosome assembly factors and r-proteins at LT” -- if you perform a post-hoc test between the low temperature and control/high temperature proteomes, is that what you see?

We performed a (post-hoc) t-test for between the 37°C and the 20°C triplicate samples. From the 54 ribosomal proteins and associated factors identified, 43 had an adjusted p-values below 0.05 (30 below 0.01) and those had a median fold change of 2.53 (2.51 respectively). We state this additional result in the text (page 5, lines 134-136).

We used this (log2 fold change-sorted) list of proteins to perform an additional GSEA analysis on webgestalt.org. We found that the term “Cytoplasmic Ribosomal Proteins” turned up again as the highest enriched gene set (together with “Principle Pathways of Carbon Metabolism”) with a p-value and FDR of 0 (aka below 1E-7), confirming again with this independent analysis the results presented in the manuscript.

(2) The FRET experiments are interesting, but seem to reach a conclusion not fully supported by the data, namely that Puf6 replaces Mg²⁺ in assisting rRNA folding (lines 364-366). Perhaps the text is just not explicitly clear here, but it currently reads as though the authors are suggesting that this is an in vivo mechanism. Were these experiments performed in cells or in vitro, or cell lysate in a tube? What impact would low temperature have on magnesium levels in cells, and therefore require replacement of Mg²⁺ with Puf6 for proper rRNA folding? This section seems promising but I couldn't quite follow the logic within the context of the larger message.

CRAC analyses and yeast three-hybrid studies indicate that Puf6 binds to the tip region of H68 within 25S rRNA. In an early nucleolar 60S pre-ribosome, H22 rRNA does not engage with H88 rRNA to form the KL, as the binding site of H22 is occluded by Ebp2 and Brx1 (Fig. 6b, State3 Nsa1/Nop2 particle). Therefore, we wondered whether Puf6 prevents H68 from forming the tertiary contact with the H22:H88 KL. To test this, we developed a minimal 65 nucleotide RNA system that recapitulates the tertiary contact between H68 TL and H22:H88 KL. The rational of the in vitro ensemble

FRET experiments are clearly described on page 11, lines 319-325. Cryo-EM studies (Sanghai et al., 2018; Kater et al., 2020; Wu et al., 2016; Wilson et al., 2020) indicate that these KL-TL interactions are developed on a compacted state of the 60S pre-ribosome (Fig. 6b, NE2 particle and Nog2 particle), but not an open extended arch-like structure seen in State 3 Nsa1/Nop2 particle. Therefore, we believe that this minimal RNA system has merit to test our hypothesis.

Minor points:

- Section “Puf6 removal from the 60S pre-ribosome precedes Nog2 recruitment” would benefit from a very brief reminder of why these particular proteins (Ssf1, Rix1, Arx1, Kre35) were chosen as baits for affinity purification in previous section “Puf6 co-enriches with early nucleolar 60S pre-ribosomes”.

The description of the maturing stages of the 60S pre-ribosome isolated via the different baits is described on page 7, lines 191-194, and a brief reminder provided on page 10, lines 286-292.

- Section on FRET experiments (“Puf6 promotes an unusual tertiary contact within the 60S pre-ribosome”) may benefit from clarification or rewording when explicitly describing the known role of (lines 314-316) to establish why these cations were chosen for the FRET studies.

RNA structure and folding is sensitive to mono- and divalent cations. For the general reader, we summarise the contribution of K^+ and Mg^{2+} ions to RNA folding and tertiary contact formation on page 11, lines 319-325.

Reviewer #3 (Remarks to the Author):

This manuscript identifies proteomic changes in yeast grown at different temperatures. The authors find that the expression levels of many ribosome assembly factors and ribosomal proteins increase under low temperature growth conditions. The classification of groups of proteins is quite interesting. The authors focus deeply on the function of one protein, Puf6, whose expression levels are high when yeast are grown at low temperature. The authors build a compelling story demonstrating that Puf6 binds to H68 in the 25S rRNA and assists its interaction with a kissing loop formed between H88 and H22. The combination of in vivo and in vitro experiments is well presented and thorough. I have only a few minor comments that can be addressed by editing the manuscript. I also suggest a couple of seemingly simple experiments that might make the story more complete, but they are not necessary to solidify conclusions. I support publication of this manuscript in Nature Communications, believing it will be of broad interest and impactful.

We thank the reviewers for her/his helpful comments, especially pointing out incorrect statements regarding Puf6-Pumilio repeat domain organization.

Suggested experiments:

Test single point mutants R172E and Y208E for function.

We have included growth analyses of the R172E and Y208E single mutants, as well as three further mutants that we had analyzed (Supplementary Fig. 4b, c, d). All substitutions map to the Puf6 surface which binds to the dsDNA substrate. However, we found that none of the single mutants were growth impaired at 20°C. Notably, Puf6-R172,431E and Puf6-Y208,431E, but not other double mutants were growth impaired at 20°C. Hence, these were used for functional studies (Fig. 4 and Fig. 5b, d).

Test Puf6 point mutants and human PUM3 in RNA-binding assays (yeast 3-hybrid and/or FP).

We have tested the interaction between Puf6-double mutants and H68 rRNA by yeast three hybrid (Fig. 5b). In contrast to WT-Puf6, both Puf6-double mutants do not interact with H68 rRNA. Together with in vivo data (Fig. 4f) and in vitro reconstitution studies (Fig. 5d) these three-hybrid studies support the idea that the interaction with H68 involves the PUF repeat domain. Unfortunately, we were unable to test interactions of PUM3 with H68 rRNA by yeast three hybrid: in contrast to pACT2-Puf6 construct, the pACT2-PUM3 construct stimulated expression of HIS3 reporter (self-activation) in absence of the H68 rRNA (see below).

Legend: The L40-coat yeast strain was transformed with pACT2-Pum3 or pACT2-Puf6 and p3A-MS2 vectors expressing empty vector or helix 68 of 25S rRNA fused to MS2 RNA. Transformants were spotted in 10-fold dilution steps on selective SD-Leu-Ura and SD-His plates containing 1 mM 3-AT and grown for 3 days at 30°C.

Lines 155-156, 212, 421-424: The description of the Pumilio repeat domain of Puf6 is not quite accurate and should cite reference #48 and Zhang, et al. 2012 (<https://link.springer.com/article/10.1007/s11033-011-0785-3>) that show the divergent Pumilio repeats in this class of PUF proteins. Other than structural similarity, the

Pumilio repeat domain in Puf6 is not the same as the well-characterized classical PUF proteins (yeast Puf3/Puf4/Puf5). As a result, Fig. 4a is a little misleading.

We thank the Reviewer for pointing this out. Puf6 is an outlier in the yeast Puf family and has a different domain organization than the other members. We revised the Fig. 4a to reflect the unique domain organization, have included references and have worked this into the text (page 7-8, lines 203-210; 14 and 15).

It is important to note that the RNA recognition properties of this domain in Puf6 are not equivalent to that of the eight-repeat single-stranded RNA-binding domains. Many of the Pumilio repeats in Puf6 lack the typical RNA base recognition side chains, especially in the C-terminal eight repeats that appear similar in shape to the single-stranded RNA-binding PUF proteins. These C-terminal eight repeats should not be referred to as canonical PUF repeats. Surprisingly, the N-terminal repeat (N-R1) is most similar to repeats found in Pumilio or yeast Puf3/Puf4/Puf5. The study here presents some important new information that could shed light on the mysterious nucleic acid recognition by Puf6. The authors could discuss this. **Do they think Puf6 recognizes the sequence or structure of H68?**

The specific crosslink between U2217 and Puf6 (Fig. 5a; CRAC analyses) and the yeast three-hybrid studies (Fig. 5f) support the notion that Puf6 binds to the tip region of H68 rRNA just below the GAAA tetraloop. Puf6 does not bind to H38 and H66 that contain a GAAA tetraloop, therefore this motif is unlikely to be the interaction platform. How Puf6 specifically recognizes H68 stem region still remains mysterious.

Lines 215-216: This section should include a reference to the small-angle X-ray scattering data in reference #48 that indicated a similar fold for yeast Puf6 compared to the crystal structure of human PUM3.

Thank you for pointing this out. We refer to the SAXS data that indicate a similar fold for yeast Puf6 and human PUM3 (page 8, lines 208-210).

Lines 216-220: It would be helpful to include the equivalent residues in the human PUM3 protein in the figure legend. The authors should also point out that the mutated residues are in the RNA-binding motifs of Pumilio repeats and are near a DNA molecule that was crystallized with human PUM3. R172 was part of a cluster of residues that were mutated previously and shown to be important for growth rate, nucleic acid interaction in vitro as well as 7S pre-rRNA processing and ASH1 localization in vivo. Y208 is also part of the same patch, and K207 was mutated. R431 was mutated as part of a patch that showed no defects. It suggests that the single mutations might be sufficient to disrupt Puf6 activity. Did the authors test the single mutations? These results should be discussed relative to the published mutation patches as they further define critical residues.

We included the corresponding residues for the human Pum3 sequence are stated in text (page 8, 214-217) and in the Figure Legend (page 32, 1041-1042).

We substituted several residues in Puf6 to glutamic acid to induce repulsion with the phosphate backbone of the RNA substrate, and thereby impair Puf6:RNA interactions (Supplementary Fig. 4b,c,d). We have included growth analyses of the R172E and Y208E single mutants, and three further mutants in Supplementary Fig. 4b,c,d. None of the single mutants were growth impaired at 20°C. Only double mutants Puf6-R172,431E and Puf6-Y208,431E were growth impaired at 20°C and consequently showed a 60S export defect (Fig. 4d, 4e). This information is included in the text on page 8, lines 217-219.

Lines 247-248, 304-308: Please include specific information about what nucleotides were included in the RNAs tested in the yeast 3-hybrid assays. This is particularly important because the authors report binding to H68 by yeast 3-hybrid assay (Fig. 5b and f), but binding to H68 was weak by FP (Fig. 6g). The sequence is provided for the FRET and FP assays, which indicates a minimal model RNA was created. This should be fully described in the text and an explanation of its validation is needed.

We have indicated the exact nucleotides that have been used in the text and have also updated Supplementary Table S4 with the plasmid sequences for the Y3H plasmids with the corresponding sequences and Supplementary Table 5 with fluorescently labeled constructs used in FP.

We have generated our minimal FRET constructs, based on the Y3H data, showing that the very tip of H68 (TL_{GAAA}) is sufficient for a stable interaction with Puf6. We have clarified this in the results section. We have also updated Fig. 6a indicating the connectivity of the individual helices by the poly-A-linker.

It is correct that pre-existing knowledge of the secondary structure was used for generating the model. This has now been rephrased in the Materials and Methods (page 26), and the 3D structure prediction of the RNA sequences is now described in the Supplementary information (page 13).

In Figure 6a (right panel), it would be helpful to indicate the polyA linkers by dashed lines to see where the connections are.

The linker has now been shown with dashed lines.

In addition, it does not seem quite right to call it a de novo model, because it uses secondary structure information in the prediction. It would be clearer to include a brief description of the method in the figure legend. The term “knowledge-based structure prediction” in Suppl. Table 5 is a good description.

A brief description of the method for “knowledge-based structure prediction is now provided in the Figure Legend.

Please present the data in Fig. 6g in the Results, not the Discussion.

The data has been presented on page 13, lines 376-380 in the Results section.

Lines 257-261: It might be helpful to list the specific crosslinks in the manuscript text or to add a title above the table, Protein:Protein Crosslinks.

A title has been added to the table in Fig. 5e.

The authors should also discuss the resolution of the crosslink data (how close residues need to be for crosslinking). It appears that K234 and K251 in uL2 are relatively close together, but K119 and K168 in Puf6 are near the N- and C-termini of the Pumilio repeat domain and quite far apart.

A line stating the resolution (range) of the cross-linking reagent has been included on page 9, line 258 and supported by a corresponding reference.

A reference to Suppl. Table 2 is needed.

This reference has been provided page 9, line 260.

Lines 1153-1154: It is probably better to call it the Pumilio repeat domain rather than the Pumilio domain.

Yes, this is correct; we have now updated our text and Figure Legends accordingly.

Lines 1157-1158: Isn't the reason that residues 118-654 are reliably modeled because the equivalent region of human PUM3 is in the crystal structure?

Yes, this is correct. Only residues 118-654 residues of the 656 residues of Puf6, could be modelled with 100% confidence using Puf-A crystal structure (pdb 4WZW) as preferred template. We have stated this in the Figure Legend page 32, 1033-1035.

Lines 1198-1201: This summary of the conclusion is confusing to me. Do you mean that the combination of the protein:protein crosslinks and RNA-binding data indicate that Puf6 binds to H68 and close to the uL2 binding site?

The Reviewer is right that the combination of protein:protein cross-links (XL-MS), CRAC data and yeast three-hybrid studies together indicate that Puf6 binds to H68 and close to the uL2 binding site. We have now revised the conclusion in the text (page 10, lines 277-279).

Figure 5a: What are the asterisks? Could you indicate the nucleotides at the beginning and end of the major peak in the CRAC data and RNA drawing?

We have updated the Fig. 5a and added also the trace of deletions that have been detected using the CRAC analysis. Deletions are indicative of directly crosslinked nucleotides and cannot be sequenced during the CRAC analysis. The asterisk (*) in Fig. 5a indicate a common contaminant (H99) in the CRAC hits that has been described previously (Granneman S *et al.* 2009). This leaves only part of helix 66, helix 68 and part of helix 69 as significant hits for the Puf6 CRAC analysis within the 25S rRNA. A second hit was detected within the ITS2 region which could be a true hit due to the close proximity of the ITS2 foot region to the L1 stalk and the proposed binding site for Puf6. The main hit region in 25S covers nucleotides C2163-A2262 which we have highlighted on the rRNA secondary structure to the right.

As requested, we have updated the Figure Legend (page 33, lines 1057-1061) and Fig. 5a by indicating the beginning and end of the peaks within the CRAC data and the rRNA drawing (Fig. 5a).

We have provided the sequences of our Y3H plasmids in the Supplementary Table 4 and have uploaded the raw sequencing data onto the Gene Expression Omnibus (GEO) database (Materials and Methods, page 22, lines 667-673).

Figure 6c: The placement of the graph legend in the top right panel is crowded and it confused me. Could it be made smaller and/or moved above the graph?

We have rearranged the panel to make it easier for the reader to follow Fig. 6c. Thank you for the suggestion.

For full transparency, the full gel blots should be included as supplementary information for figures that show fragments of Western blots.

We have provided the uncropped data for Western blots as an additional attachment named Source data.

Reviewers' Comments:

Reviewer #1:

Remarks to the Author:

The authors have addressed all main suggestions raised by the three reviewers at an appropriate level. This is an interesting, impactful and convincing manuscript that I strongly recommend for publication in Nature Communications.

Ute Kothe

Reviewer #2:

Remarks to the Author:

The authors have addressed my concerns satisfactorily, and I look forward to the publication of their work in Nature Communications.

Reviewer #3:

Remarks to the Author:

The authors have addressed all of my concerns, which were minor. Thank you for being so thorough. I found a couple of small edits that could be made. I recommend publication in Nature Communications.

Page 3, lines 64-65: I think it should read, forms the "foot" structure already visible in these states."

Page 14, line 432: Suggest rewording the beginning of the sentence for clarity to "Yeast Puf6, and its human ortholog Pum3/Puf-A, adopts..."

“Puf6 primes 60S pre-ribosome nuclear export at low temperature”

(NCOMMS-21-11977A)

Point-to-Point response (in blue)

REVIEWER COMMENTS (in black)

Reviewer #1 (Remarks to the Author):

The authors have addressed all main suggestions raised by the three reviewers at an appropriate level. This is an interesting, impactful and convincing manuscript that I strongly recommend for publication in Nature Communications.

Ute Kothe

We thank Ute for her constructive suggestions that have improved our manuscript.

Reviewer #2 (Remarks to the Author):

The authors have addressed my concerns satisfactorily, and I look forward to the publication of their work in Nature Communications.

We thank this Reviewer for his/her constructive suggestions that improved our presentation of the mass spectrometry data.

Reviewer #3 (Remarks to the Author):

The authors have addressed all of my concerns, which were minor. Thank you for being so thorough. I found a couple of small edits that could be made. I recommend publication in Nature Communications.

We thank this Reviewer for shining light on PUF-protein family and helping to improve this aspect our manuscript. As requested, we have corrected the following sentences in the text.

Page 3, lines 64-65: I think it should read, forms the “foot” structure already visible in these states.”

Line 64, corrected as stated above

Page 14, line 432: Suggest rewording the beginning of the sentence for clarity to “Yeast Puf6, and its human ortholog Pum3/Puf-A, adopts...”

Line 425, corrected as stated above.